# A unified approach for quantifying and interpreting DNA shape readout by transcription factors

H Tomas Rube[1], Chaitanya Rastogi[1,2], Judith F Kribelbauer[1,3] & Harmen J Bussemaker[1,3,*] (iD)

## Abstract

Transcription factors (TFs) interpret DNA sequence by probing the chemical and structural properties of the nucleotide polymer. DNA shape is thought to enable a parsimonious representation of dependencies between nucleotide positions. Here, we propose a unified mathematical representation of the DNA sequence dependence of shape and TF binding, respectively, which simplifies and enhances analysis of shape readout. First, we demonstrate that linear models based on mononucleotide features alone account for 60–70% of the variance in minor groove width, roll, helix twist, and propeller twist. This explains why simple scoring matrices that ignore all dependencies between nucleotide positions can partially account for DNA shape readout by a TF. Adding dinucleotide features as sequence-to-shape predictors to our model, we can almost perfectly explain the shape parameters. Building on this observation, we developed a *post hoc* analysis method that can be used to analyze any mechanism-agnostic protein–DNA binding model in terms of shape readout. Our insights provide an alternative strategy for using DNA shape information to enhance our understanding of how *cis*-regulatory codes are interpreted by the cellular machinery.

**Keywords** DNA binding specificity; DNA shape; sequence-readout mechanisms; statistical analysis; transcription factors
**Subject Categories** Chromatin, Epigenetics, Genomics & Functional Genomics; Methods & Resources; Transcription
**Mol Syst Biol. (2017) 14: e7902**

## Introduction

A central goal of regulatory genomics research is to understand how transcription factors (TFs) are recruited to specific regulatory elements. Popular approaches to quantifying DNA binding by a particular TF include *in vivo* profiling using ChIP-seq (Stormo, 2000; Foat *et al*, 2006) and high-throughput *in vitro* assays (Matys *et al*, 2003; Mukherjee *et al*, 2004; Maerkl & Quake, 2007; Slattery *et al*, 2011; Weirauch *et al*, 2014). DNA binding specificity is often

represented in terms of a scoring matrix [also referred to as "position-specific scoring matrix", "position weight matrix" (Stormo, 2000), or "position-specific affinity matrix" (Foat *et al*, 2006)], under the assumption that each base pair position within the binding site contributes independently to the overall affinity. Scoring matrices for thousands of TFs are available in online databases (Robasky & Bulyk, 2011; Mathelier *et al*, 2014). While many algorithms for constructing scoring matrices exist, the most accurate of these are based on biophysical models of protein–DNA interaction that take dependencies between neighboring nucleotide positions into account (Weirauch *et al*, 2013; Riley *et al*, 2015).

Having a scoring matrix available for a particular TF does not mean that the structural mechanisms the TF employs to bind more strongly to some DNA sequences than to others are known. This is not a problem when the goal is to predict the landscape of (relative) binding affinity along the genome or the impact of non-coding polymorphisms on TF binding. However, to understand or predict the impact of amino acid substitutions in the DNA binding domain of a TF (Abe *et al*, 2015; Barrera *et al*, 2016), structural insight is indispensable.

In this study, we present an approach for analyzing a mechanism-agnostic model (such as a scoring matrix) in order to reveal the readout mechanisms it implicitly encodes. A classic and well-understood mechanism is base readout, where TF residues form stabilizing bonds with specific bases or pairs of stacked bases in the major groove (Luscombe *et al*, 2001). A complementary mechanism called DNA shape readout, defined as sensitivity of TF binding to subtle deviations from average B-DNA helical structure (Hegde *et al*, 1992; Parkinson *et al*, 1996; Hizver *et al*, 2001), has received significant attention in recent years (Rohs *et al*, 2009, 2010; Slattery *et al*, 2011; Zhou *et al*, 2015). In particular, for the *Drosophila* Hox protein *Sex combs reduced* (Scr) in complex with *Extradenticle* (Exd), crystallographic analysis of binding to high- and low-affinity DNA ligands (Rohs *et al*, 2010), as well as comparison between the binding specificity of Exd-Scr and that of complexes with a mutated or paralogous Hox protein (Slattery *et al*, 2011; Abe *et al*, 2015), implicates a specific arginine residue in the N-terminal side chain of Exd in mediating minor groove width (MGW) readout.

While these structural studies demonstrate how shape readout can be identified and rigorously validated for specific cases, the

1 Department of Biological Sciences, Columbia University, New York, NY, USA
2 Program in Applied Physics and Applied Mathematics, Columbia University, New York, NY, USA
3 Department of Systems Biology, Columbia University Medical Center, New York, NY, USA
*Corresponding author. Tel: +1 212 854 9932; E-mail: hjb2004@columbia.edu

experimental methods used are both costly and time-consuming. Statistical methods for detecting signatures of shape readout from *in vitro* TF binding specificity data alone have therefore been developed (Zhou *et al*, 2015; Yang *et al*, 2014, 2017; Abe *et al*, 2015). Detecting DNA shape readout depends critically on knowing the detailed shape of potential TF binding sites. The conformation of the DNA helix is parameterized using two main classes of parameters: (i) base pair parameters that specify the relative location of the bases in a base pair, and (ii) base pair step parameters that specify the relative location of two adjacent base pairs (Olson *et al*, 2001). Together, these parameters fully specify the location of the bases. For B-DNA, some of the most important are the base pair parameter Propeller Twist (ProT), and step parameters Slide, Roll, and Helix Twist (HelT; Calladine & Drew, 1992). Derived quantities that are more closely related to specific readout mechanisms, such as MGW and electrostatic potential (Rohs *et al*, 2009), are also used. Studying the sequence dependence of DNA shape and deformability has a long history (Calladine, 1982; Calladine & Drew, 1992). While early studies used crystal structure data to tabulate the shape and flexibility of dinucleotide steps (Olson *et al*, 1998), a more recent study used Monte Carlo simulations to predict HelT, Roll, ProT, and MGW in all pentameric sequence contexts (Zhou *et al*, 2013). Unlimited by the availability of experimental structures, these tables have proven highly useful, as the shape profile along any DNA sequence can be predicted quickly using a simple sliding window.

Recent years have seen flurry of computational work relating to DNA shape readout (Gordan *et al*, 2013; Zhou *et al*, 2015; Yang & Ramsey, 2015; Mathelier *et al*, 2016; Ma *et al*, 2017). One of two main approaches is typically utilized. The first is to contrast the shape of high- and low-affinity binding sites and interpret the observed differences as evidence for shape readout (Joshi *et al*, 2007; Gordan *et al*, 2013; Zhou *et al*, 2013). Methods for

assessing the statistical significance of such differences, however, have not yet been developed. The second approach is to use DNA shape features as predictors in mathematical models of TF binding. The rationale is that shape readout gives rise to specific dependencies between nucleotide positions that cannot be captured by a simple scoring matrix. Adding DNA shape as a predictor to TF binding models based on mononucleotide features alone indeed improves to ability to predict TF binding *in vivo* and *in vitro* (Zhou *et al*, 2015; Mathelier *et al*, 2016), as well as the ability to predict gene expression (Peng & Sinha, 2016). However, this improvement is similar to that observed when di- and trinucleotide features instead are added as predictors to the binding models. This equivalence has been interpreted to reflect that dinucleotides capture base-stacking interactions and trinucleotides capture short structural elements such as A-tracts (Zhou *et al*, 2015; Yang *et al*, 2017). However, it has not been systematically explored to what extent the variation in the pentamer-based shape tables can be explained in terms of effects associated with simpler DNA sequence features such as mono- and dinucleotides.

Identifying DNA shape readout when TF binding specificity data are available requires one to dissect specificity in terms of base and shape readout (Fig 1A–I). Base readout (Fig 1D) is commonly parameterized using a scoring matrix (Fig 1G), and the corresponding contribution to the binding affinity is computed by combining this matrix with base identity indicators (Fig 1A) along the binding site (we reserve the term *scoring matrix* for TF binding models that only have mononucleotide predictors). Shape readout (Fig 1F) is parameterized using a profile of shape-sensitivity coefficients along the protein–DNA interface (Fig 1I), which have the interpretation of change in (normalized) binding free energy ($\Delta\Delta G/RT$) per unit (angular degree for Roll, HelT, ProT; Ångström for MGW) of change in shape-parameter value. The corresponding affinity contribution can

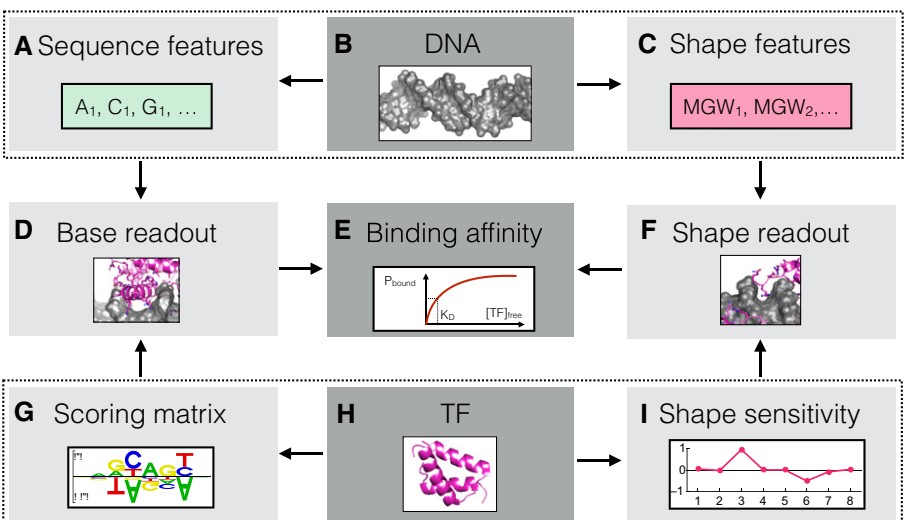

**Figure 1. Quantifying base and shape readout.**

A–I A transcription factor (TF) (H) recognizes a DNA molecule (B) either through base readout (D) or through readout of other biophysical characteristics such as DNA shape (F). Base readout is quantified using DNA sequence feature indicators (A) and a scoring matrix (G). Similarly, shape readout is quantified in terms of shape parameters (C) and shape-sensitivity coefficients (I) along the DNA molecule. Structures for PDB/2R5Y were generated using PyMOL.

be computed by multiplying these coefficients with the shape parameters along the binding site (Fig 1C; Zhou *et al*, 2015). Since sequence determines shape, shape sensitivity at a specific position should be equivalent to readout of a specific combination of (possibly higher-order) sequence features. However, representing DNA shape using pentamers, as is the current standard, makes it difficult to see exactly what these equivalent sequence features are. A more parsimonious sequence-to-shape model with fewer parameters could in turn facilitate analysis of the relationship between shape readout and binding specificity and, ideally, allow for statistical hypothesis testing based on specific signatures of DNA shape readout.

In the present study, we revisit the problem of attributing DNA shape readout and address a number of open questions. We start by asking what the distinctive signatures are of shape readout and how different they are from the signatures of base readout. Based on a simple analysis, we show that standard scoring matrices that ignore any dependencies between nucleotide positions within the binding site can capture shape readout to a surprising extent. Prompted by this observation, we show that the sequence-to-shape relationship encoded in commonly used pentamer-based shape tables can be almost perfectly parameterized in terms of mononucleotide and dinucleotide features. We next develop a practical two-step approach in which quantification of DNA binding specificity is based on a strictly mechanism-agnostic model, and interpretation in terms of shape readout is implemented as a *post hoc* analysis of this model. We show that this approach prevents numerical instabilities that hinder the direct interpretation of shape-sensitivity coefficients in hybrid models in which sequence and shape features compete to explain binding data. Importantly, our approach also provides a natural way to assign statistical significance to the readout of a particular shape parameter at a particular position within the binding site.

# Results

### Scoring matrices encode known associations between DNA shape and binding affinity

Whenever binding affinity has been quantified for a large number of DNA sequences in a high-throughput assay, and these sequences can somehow be aligned with each other, one can look for associations between binding and shape that are indicative of shape readout. Indeed, several studies have compared the shape profile for high-affinity TF binding sites (determined either using the crystal structure of protein–DNA complexes or by computing the shape of individual high-affinity sequences using the pentamer tables) with that of lower-affinity sites (Joshi *et al*, 2007; Slattery *et al*, 2011; Zhou *et al*, 2013; Shazman *et al*, 2014; Yang *et al*, 2014), or looked for correlations between binding and shape (Gordan *et al*, 2013). In particular, the binding specificity of the Exd-Scr complex was previously determined using SELEX-seq, a high-throughput *in vitro* assay (Slattery *et al*, 2011); it was found that high-affinity binding sites have a shape where the minor groove is narrowed in the center of both the Exd and the Scr half-site (Fig EV1A).

When a scoring matrix is available for a particular TF, it can be used to rank sequences by binding affinity. This made us wonder to what extent associations between DNA shape and binding affinity might already be encoded in a scoring matrix (which completely neglects dinucleotide and higher-order dependencies). To answer this question, we fit a simple free-energy model with additive mononucleotide effects only to the published SELEX-seq data for Exd-Scr (Materials and Methods; Fig 2A); this was done using a novel algorithm named *No Read Left Behind* (NRLB), which maximizes the likelihood of the entire set of SELEX probes sampled after one round of affinity-based selection (C. Rastogi, H. T. Rube, J. F. Kribelbauer, J. Crocker, R. E. Loker, G. D. Martini, O. Laptenko, W. Freed-Pastor, C. L. Prives, D. L. Stern, R. S. Mann, H. J. Bussemaker, in preparation; see Materials and Methods for details).

To see what associations between DNA shape and binding affinity are encoded in the *NRLB* mononucleotide model for Exd-Scr, we randomly sampled DNA sequences, computed their binding affinity, segregated the sequences into affinity bins, and computed the average MGW profile for each bin using the MGW pentamer table (Fig 2B). This revealed that the binding sites identified by the scoring matrix had a narrowed minor groove in the center of both the Exd and the Scr half-site, consistent with the original analysis (Slattery *et al*, 2011), which was based on oligomer count tables (Fig EV1A). Repeating the same analysis for ProT, Roll, and HelT also gave good agreement with the average shape profile of a set of high-affinity SELEX probes aligned based on their sequence. SELEX-seq data are also available for the Hox factor *Ultrabithorax* (UbxIVa) in complex with Exd, a complex that does not select a narrow groove in the Hox half-site but instead prefers it to be slightly widened. Repeating the analysis for this dataset indeed revealed an increased MGW in the UbxIVa half-site (Figs 2C and D, and EV1B). We also reanalyzed SELEX-seq data for MAX and found good agreement (Figs 2E and F, and EV1C).

A large body of publicly available TF binding data has also been generated using protein binding microarray (PBM) technology (Mukherjee *et al*, 2004). A recent study reanalyzed PBM data curated in the JASPAR (Mathelier *et al*, 2014) and UniPROBE (Robasky & Bulyk, 2011) databases and computed the average DNA shape profile of high-affinity sequences for a large number of transcription factors (Yang *et al*, 2014). To see whether these profiles can be derived using PBM-based scoring matrices, we arbitrarily focused on the nuclear receptor HNF4A, downloaded the scoring matrix and binding PBM probe sequences for from UniPROBE, repeated the above analysis, and found that the scoring matrix (Figs 2G and H, and EV1D) indeed encodes the previously reported mean shape of high-affinity probes (Yang *et al*, 2014). We also downloaded PBM data for the arbitrarily chosen *Drosophila* nuclear hormone receptor Ftz-F1 from CIS-BP (Weirauch *et al*, 2014), repeated the analysis, and found good agreement (Figs 2I and J, and EV1E). Thus, our approach is generally useful across multiple high-throughput *in vitro* profiling platforms. Altogether, these results show that while SELEX-seq and PBM can identify preferences for complex higher-order sequence features, a simple scoring matrix somehow already encodes known differences in DNA shape between high- and low-affinity binding sequences.

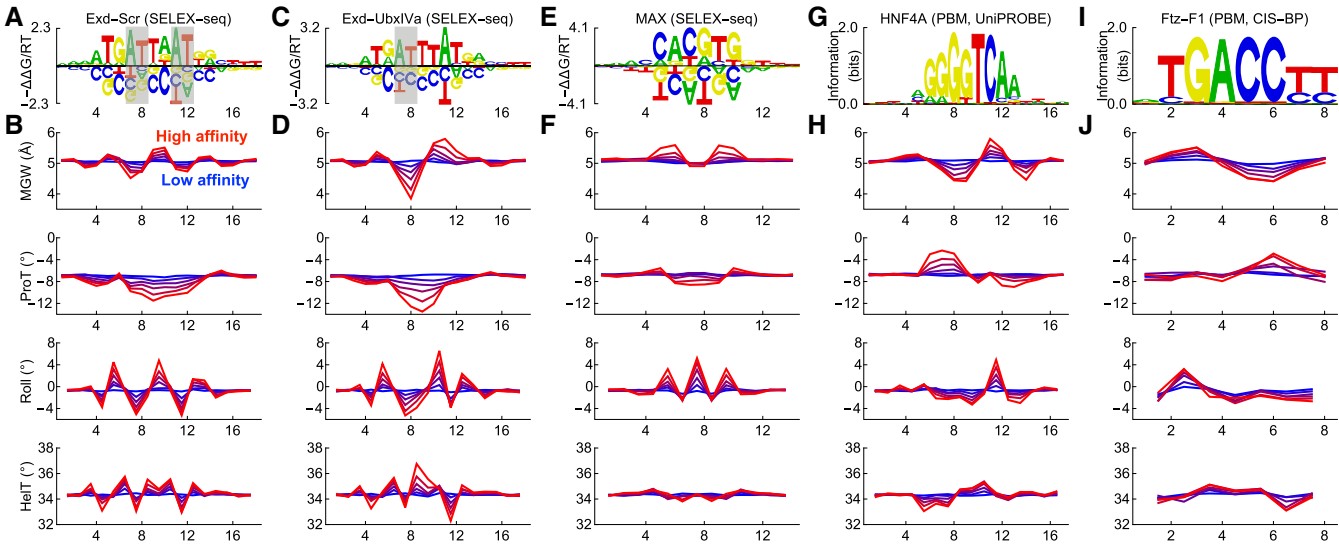

**Figure 2.  Scoring matrices encode associations between DNA shape and binding affinity.**

A   Free-energy scoring matrix for the Exd-Scr complex derived from SELEX–seq data using NRLB. Letter height indicates the magnitude of the position-specific free-energy contribution, and inverted lower letters indicate destabilizing bases (Foat *et al*, 2006). Shaded rectangles indicate known sites of MGW readout.

B   Mean MGW, ProT, Roll, and HelT profiles of Exd-Scr binding sites identified using the scoring matrix in (A), stratified by affinity. Red and blue lines indicate high- and low-affinity sites, respectively. The profiles were generated by drawing random 18mer sequences, computing the free energy of the sequences using the scoring matrix, sorting them into free-energy bins of equal size, and finally computing the average shape of the sequences in each bin.

C, D   The same as (A, B) but for the Exd-UbxIVa complex.

E, F   The same as (A, B) but for the MAX homodimer complex (Zhou *et al*, 2015).

G, H   The same as (A, B) but using BPM-based scoring matrix for HNF4A downloaded from UniPROBE (ID: 2640).

I, J   The same as (A, B) but using BPM-based scoring matrix for Ftz-F1 downloaded from CIS-BP (ID: M1470_1.02).

## A mononucleotide model captures two-thirds of the variance in DNA shape parameters

The fact that a simple scoring matrix can parameterize a significant part of the shape readout by the Exd-Scr complex made us wonder to what extent similar models might be used to capture the information contained in the pentamer-based shape tables themselves in a more compact and interpretable manner (Fig 3A). In other words, we wished to consider the precise nature of the relationship between DNA sequence and shape in more detail. To this end, we predicted the shape parameters across all pentamers using linear regression models that only included mononucleotide features. Our models for the step parameters Roll and HelT used a 4-bp sequence window centered at the base pair step (with 7 degrees of freedom, DOF) and explained 74 and 71%, respectively, of the variance when hold-one-out cross-validation was used (Fig 3B and C). For the base pair parameter ProT and for the MGW, our models used 5-bp sequence windows centered at the base pair (with 8 DOF) and accounted for 67 and 60% of the variance, respectively (Fig 3B and C).

We used a sequence logo representation to visualize the sequence-to-shape model regression coefficients (Fig 3D). For the step features (Roll and HelT), the coefficients were largest immediately before and after the step. A model in which only these coefficients were allowed to be nonzero performed almost as well, explaining 73 and 61% of the variance, respectively; since reverse-complement symmetry relates the regression coefficients on either side of the step with each other, the sequence dependence in these reduced models can be parameterized by three independent parameters and an intercept. For ProT, the regression coefficients were largest at the central base pair. A model using only the central base pair accounted for 54% of the variance. Reverse-complement symmetry maps the central base pair to itself; the sequence dependence of this model therefore is set by as single parameter encoding the G/C dependence. G:C base pairs have less (negative) ProT than A:T base pairs, due to the additional stabilizing hydrogen bond in the former. This association has also been noted previously (Hancock *et al*, 2013; Dror *et al*, 2015).

The MGW regression coefficients were small at the central base pair but large one and two base pairs away. At each position, A and T had much larger values than G and C. The A and T coefficients had opposite signs at each base pair and (therefore, following reverse-complement symmetry) opposite signs on the 5′ and 3′ sides of the central position. The shape of A-tracts has been studied extensively, and our MGW model recapitulated earlier observations (Haran & Mohanty, 2009; Rohs *et al*, 2010); the $A_4T_4$ octamer is curved and has narrowed minor groove around the ApT step, whereas the $T_4A_4$ octamer is straight and has wider minor groove around the TpA step (Burkhoff & Tullius, 1987, 1988; Haran & Mohanty, 2009). The difference in curvature between these two octamers has been attributed to a negative Roll in the ApT step and a positive Roll in the TpA step (Stefl *et al*, 2004), both of which can be seen in the Roll model. The minor groove of A-tracts is progressively narrowed in the 5′-to-3′ direction (Burkhoff & Tullius, 1987, 1988), and this is recapitulated by applying our MGW model across sequences of type

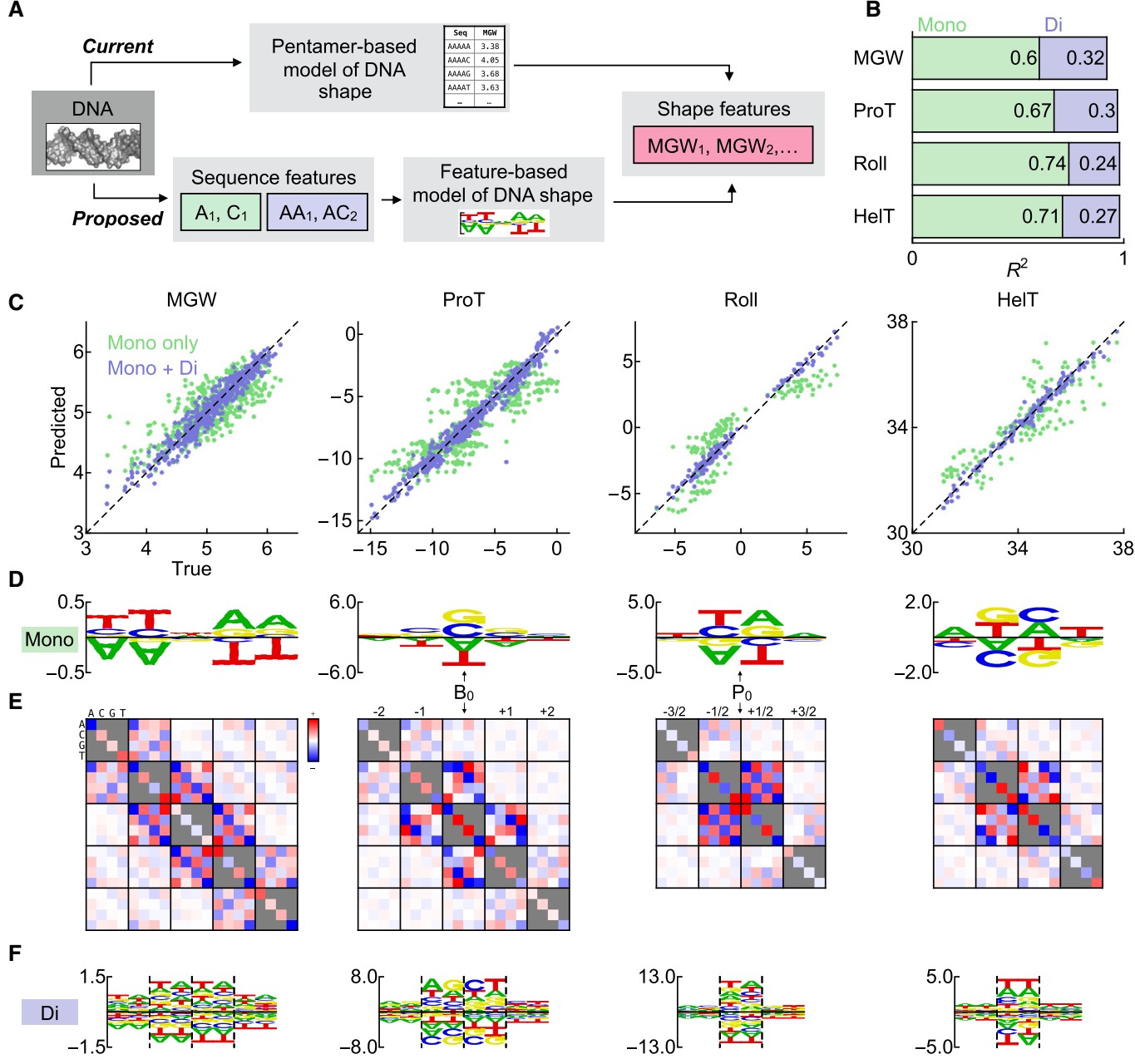

**Figure 3. Mononucleotide models describe sequence-to-shape relationships well.**

A   Schematic diagram showing how pentamer tables currently are used to evaluate the DNA shape parameters and how we alternatively evaluate them using sequence feature-based models.

B   Fraction of variance explained ($R^2$) for models with mononucleotide predictors only (green) and models with mono- and dinucleotide predictors (increase shown in blue). Leave-one-out cross-validation was used.

C   Scatter plots of true vs. predicted shape parameters. Green dots indicate predictions by the mononucleotide models, whereas blue dots indicate predictions by models that also include dinucleotide interactions.

D   Regression coefficients of mononucleotide-only sequence-to-shape models, shown as logos. The models were trained on the shape pentamer tables of Zhou *et al* (2013). $B_0$ and $P_0$ indicate the positions of the central base and base pair step, respectively.

E   Regression coefficients for an all-by-all interaction model (red, above average; blue, below). Diagonal blocks indicate self-interactions (equivalent to mononucleotide-based predictors), and off-diagonal blocks indicate interactions.

F   Logo representation of dinucleotide interactions using the representation in (D). Dashed lines separate columns of dinucleotides.

$N_2A_5N_2$. The base pairs in A-tracts are also strongly propeller-twisted (Nelson *et al*, 1987); this is recapitulated by the large (negative) values for A/T bases in the ProT model.

Visualizing the sequence dependence encoded in the pentamer tables as sequence logos clarifies how they are reflected in the TF scoring matrices. For concreteness, consider the ProT profile of

Exd-Scr (Fig 2B) and note how the peaks and troughs align with the G/C preferences in the scoring matrix. The ProT profile may look like an impressive signature for shape readout, but it should be kept in mind that other mechanisms such as direct base recognition may also confer a preference for G/C nucleotides. This example illustrates how inspecting the sequence-to-shape model logos in Fig 3 can help judge signatures of shape readout.

## Dinucleotide features predict shape parameters with high precision

While dependency on mononucleotide features alone captured most of the variance in the shape tables as discussed above, the neglected higher-order sequence dependence is thought to also be biophysically important. Indeed, including shape features as predictors can boost the performance of TF binding models, suggesting that shape and mononucleotide features at least partly complement each other (Abe *et al*, 2015; Zhou *et al*, 2015). Models that include di- and trinucleotide rather than shape features as additional predictors have roughly similar performance (Zhou *et al*, 2015; Yang *et al*, 2017). Together, these observations have led to the notion that shape features mostly encode these higher-order sequence features. However, to what extent dinucleotide features help explain variation in the shape features themselves has not, to our knowledge, been directly addressed.

To investigate the nature of the higher-order sequence dependencies in the shape table, we used multiple linear regression to fit sequence-to-shape models that included all pairwise interactions between nucleotide positions, adjacent and non-adjacent. These models performed even better than the mononucleotide-only models, respectively explaining 93, 97, 99, and 98% of the variance for MGW, ProT, Roll, and HelT. Repeating the analysis with permuted tables gave models explaining less than 1% of the variance, showing that the performance of the models is not due to overfitting. The regression coefficients were on average 6.2-fold larger for interactions between neighboring nucleotides than for interactions between non-adjacent positions (Fig 3E), indicating that these dinucleotide features are the most important. Indeed, model performance barely degraded when only dinucleotide features were included (explaining 92, 97, 99, and 98%, respectively, of the variance for MGW, ProT, Roll, and HelT). These models had 26 and 22 DOF for the base and step parameters, respectively. For the step parameters Roll and HelT, the interaction coefficients were small outside base pair steps (Fig 3F). TT and AA interactions in the ProT model increase the (negative) twisting angle, an effect due to a cross-strand hydrogen bond (Nelson *et al*, 1987; Calladine & Drew, 1992). Dinucleotide interactions also increased the Roll angle for the pyrimidine–purine step, an effect due to a cross-chain purine clash (Calladine & Drew, 1992). While the mononucleotide MGW model had small coefficients for the central nucleotide, the dinucleotide interactions for the central base were quite large. Including these interactions further captured narrowing of the minor groove of A-tracts and widening for the TpA step.

## Direct modeling of shape readout

We now turn to the problem of identifying shape readout by examining the sequence-to-affinity relationship of a TF. Biophysically, we expect shape readout to be manifested in this relationship as a dependence on the shape parameters that is independent of the base readout (cf. Fig 1). However, because the shape cannot vary without changing the base sequence, shape and base readout must be treated in a unified manner. We therefore model the TF binding energy using both sequence and shape features (Fig 4A): Binding energy is computed by (i) scoring the sequence using a scoring matrix which encodes base readout, and (ii) computing the shape parameters along the binding site and multiplying these values by corresponding sensitivity coefficients that encode shape readout. After fitting such a model to the TF binding data, we may be able to identify shape readout either by examining the inferred shape-sensitivity coefficients or by comparing the performance of this model to a model that does not include shape readout. Because the binding model is fit directly to the data, we call this approach *direct shape regression*.

Models of TF binding specificity utilizing linear combination of base and shape features have been used previously in the literature, including a study that used linear support vector regression to model the logarithmic PBM probe intensity (Zhou *et al*, 2015) and a study that used $L_2$-penalized multiple linear regression to model the binding affinity inferred from HT-SELEX data (Yang *et al*, 2017). Both these studies found that including shape features as predictors improves the performance of models of *in vitro* binding.

Revisiting the Exd-Hox SELEX-seq data, we used *NRLB* (C. Rastogi, H. T. Rube, J. F. Kribelbauer, J. Crocker, R. E. Loker, G. D. Martini, O. Laptenko, W. Freed-Pastor, C. L. Prives, D. L. Stern, R. S. Mann, H. J. Bussemaker, in preparation) to fit binding models that included both mononucleotide and shape features as predictors of binding free energy (see Fig 4A and Materials and Methods). We quantified model performance by first tabulating the occurrence of 10mers in the SELEX-seq library by sliding a window across each probe (for SELEX round one), then computing the 10mer frequencies predicted by the models, and finally using the Pearson correlation coefficient of the log-transformed counts to compare the observed and predicted values (Fig 4B and C). Comparing the mononucleotide-only model for Exd-Scr (cf. Fig 2A) to a direct shape regression model that included MGW as an additional predictor, we found that predictive performance increased modestly but statistically significantly ($r^2 = 0.752$ vs. $r^2 = 0.789$, $P = 1 \times 10^{-16}$, *z*-test; see Materials and Methods), consistent with the prior observation that DNA shape parameters are useful for TF binding prediction (Zhou *et al*, 2015). Repeating the analysis using the Pearson and Spearman correlation coefficients for the (non-transformed) counts gave similar results (Fig EV2A and B).

What implications does the fact that mono- and dinucleotide sequence features together predict the shape parameters almost perfectly have on the problem of identifying DNA shape readout? Earlier studies compared the predictive power of the DNA shape parameters (MGW, ProT, HelT, and Roll, together with four quadratic interactions between adjacent parameter values) to the predictive power of di- and trinucleotides (Zhou *et al*, 2015; Yang *et al*, 2017). Since these two sets of parameters have similar predictive power, it was previously conjectured that the reason why di- and trinucleotides partially capture readout is that dinucleotides describe stacking interactions and trinucleotides describe short structural elements such as A-tracts. To explicitly test whether the predictive power of the shape parameters is fully encapsulated by dinucleotide features, we repeated the *NRLB* fit using the shape parameters predicted by the dinucleotide-based sequence-to-MGW model (cf. Fig 3F), thereby ignoring all

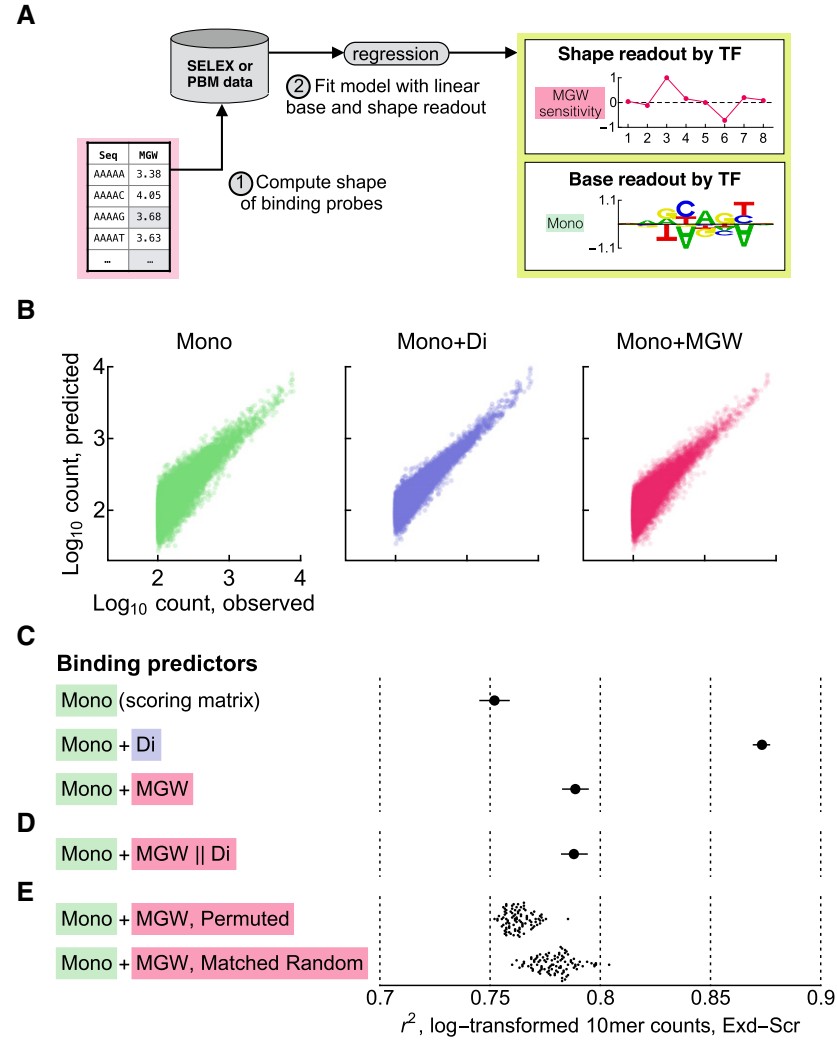

**Figure 4.  Analyzing DNA shape readout through direct shape regression.**

A   Schematic diagram showing the workflow of direct shape regression. The shape of individual DNA probes is first determined using the standard pentamer tables. Next, a protein–DNA recognition model that uses both mononucleotide and shape features along the probes as predictors (as in Fig 1) is fit to the binding data.

B   Scatter plots showing observed vs. predicted 10mer counts for Exd-Scr binding models with different sets of predictors. The predictor combinations used were as follows: mononucleotide only (cf. scoring matrix in Fig 2A); mono- and dinucleotide (mechanism-agnostic readout model); and mononucleotide and MGW from pentamer table. 10mers with count 100 or greater are shown.

C   Performance of the binding models in (B) as quantified by the Pearson correlation ($r^2$) between the logarithm-transformed counts. Error bars show 95% confidence interval computed using the Fisher $z$-transformation. 50% of the reads in SELEX round one were held out for this validation.

D   The same as (C) but with the mononucleotides and the dinucleotide component of MGW (cf. Fig 3F) as predictors.

E   The same as (C) but with the mononucleotides and 100 randomly permuted MGW pentamer tables (top) or 100 complexity-matched random pentamer tables (bottom, cf. Materials and Methods and Fig EV4B) as predictors.

possible higher-order contributions. The resulting model performs virtually identically to that obtained using the original MGW table ($r^2 = 0.789$, $P = 0.788$; Fig 4D), consistent with the above observation that over 90% of the shape variance in the pentamer tables can be accounted for without including trinucleotide features.

**Detecting shape readout through *post hoc* analysis of mechanism-agnostic models**

The direct shape regression approach outlined above has some drawbacks. First, the shape-sensitivity coefficients will only be

nonzero insofar as the resulting dinucleotide effects complement the scoring matrix. However, we observed in Fig 3 that two-thirds of the variance in shape can be captured using mononucleotide features alone. It would therefore be preferable to also use the information in the scoring matrix to identify shape readout. Second, while including shape readout improved performance, the inferred sensitivity coefficients often appeared unstable (Fig EV3A), suggesting that the problem was poorly conditioned. Third, mechanisms other than shape readout can also contribute to dinucleotide effects in the binding specificity model. This makes it desirable to have a framework for assessing the statistical significance with which part

of the binding specificity can be attributed to readout of a specific shape feature at a specific position within the binding site.

In the following sections, we will develop an alternative two-step method for shape-readout analysis that addresses these issues by fundamentally changing the way in which the sequence-to-shape models are used: The first step is to quantify TF binding specificity using a mono- and dinucleotide binding model that is agnostic about the underlying molecular readout mechanisms. The second step is to identify the weighted combination of readout modes that best explains the mechanism-agnostic model. We will call this *post hoc* analysis step *shape projection* (Fig 5A); we implemented it as software we named *Shapelizer* (Materials and Methods). An advantage of our two-step approach is that it completely separates quantification of TF binding specificity from interpretation in terms of readout mechanism. In practical terms, this means that the *post hoc* analysis can be performed without revisiting the underlying data, for example, when additional shape parameters become available (Chiu *et al*, 2017; Li *et al*, 2017), or when assessing statistical significance by fitting a large number of readout models with parameters drawn from an appropriate null distribution (as we will do below).

### Sequence-only representation of shape readout

For the shape projection approach to be feasible, we need to know how shape readout manifests itself through the mono- and dinucleotide preferences of the mechanism-agnostic model (Fig EV4A). To illustrate this relationship, consider a TF preferring a narrow minor groove at a specific position in a binding site. Since the minor groove will be narrowed by adenines directly 5′ and thymines directly 3′ to this position (cf. sequence-to-MGW model logo in Fig 3D), the preference for a narrowed minor groove should be reflected at corresponding positions in the sequence-to-affinity model. The mechanism-agnostic model would similarly include contributions that reflect the dinucleotide interactions in the sequence-to-MGW model (cf. Fig 3F). Shape readout at additional positions, or of additional features, can be accounted for by adding corresponding values in the sequence-to-shape model at the appropriate offset and amplitude. This procedure for expressing shape readout can be formulated mathematically as a convolution between the profile of shape sensitivity along the binding site and the sequence-to-shape regression coefficient for a particular mono- or dinucleotide feature. Expressing shape readout in terms of recognition of mono- and dinucleotide features is thus straightforward. However, as we will see below, solving the inverse problem of inferring shape readout from the mechanism-agnostic mono- and dinucleotide model requires more sophisticated computational methods.

### Shape projection: from binding quantification to readout interpretation

In this section, we will solve the inverse problem of inferring shape readout from a given mechanism-agnostic model. We first define a loss function that quantifies the disagreement between the mechanism-agnostic model that was learned from the binding data and the specific shape sensitivities that we are estimating. We then optimize this loss function with respect to the latter (Fig 5A). The choice of loss function was motivated by how a TF samples binding sequences with a probability that is proportional to the binding affinity. The Kullback–Leibler (KL) divergence is an information-theoretic measure of how well an approximate probability distribution recapitulates a true distribution. We define the loss function as the KL divergence between the (approximate) base- and shape-readout model and the (true) mechanism-agnostic model.

One difficulty in distinguishing between shape and base readout is that the scoring matrix already encodes part of shape readout, as we demonstrated above. Had the mononucleotide-based sequence-to-shape models been perfect, base and shape readout would have been perfectly collinear and the inference problem ill-conditioned. In a strict sense, breaking the degeneracy between base and shape readout therefore requires knowledge about binding preferences for higher-order sequence features. For the problem at hand—where shape readout at a single base is equivalent to a specific signature of base readout across multiple bases (Fig EV4A)—we prefer parsimonious solutions where the shape and base readout both are as localized as possible and mutual cancelation is minimal. We accomplish this by adding regularization terms to the loss function (see Materials and Methods). This introduces competition between the parameters to explain the data and therefore leverages both the mono- and dinucleotide binding preference to identify shape readout.

To demonstrate that this approach is feasible, we again returned to the Exd-Scr SELEX-seq dataset, fit a mechanism-agnostic readout model with mono- and dinucleotide predictors using NRLB, and then used $L_2$-regularized shape projection to infer the shape-sensitivity coefficients (Fig 5B). This produced robust shape-sensitivity profiles with distinct preferences for narrow minor grooves in the central Exd and Scr half-sites. Repeating the analysis for the Exd-UbxIVa complex yielded sensitivity profiles that lack narrowing in the UbxIVa half-site (Fig 5C). These results are consistent with previous findings (Slattery *et al*, 2011). Our fits were relatively insensitive to the details of how the penalization was implemented and to using an alternative least-squares loss function (Fig EV6). However, dropping the penalty term from the model altogether gave inconsistent results due to the same issues of poor conditioning that plague the direct shape regression approach described above (Fig EV3A and B). We conclude that penalized shape projection procedure successfully overcomes the confounding between base and shape readout that arises due to their inherent similarity, by making the mathematical symmetries that connect them explicit.

### Statistical significance of shape-readout attribution

Modeling MGW readout clearly improves model performance, and the MGW sensitivity coefficients inferred using the projection methods discussed above agree with previously validated instances of MGW readout. However, it is not immediately obvious how statistically significant these signatures are. We therefore asked whether the observed performance boost was larger than expected had we instead added a random sequence feature. This same issue was previously addressed by generating random shape tables where the association between pentamers and shape parameters was randomly permuted (Zhou *et al*, 2015). These permuted tables did not give the same predictive advantage, suggesting that the performance increase observed for the true shape tables is biophysically relevant. We similarly generated 100 randomly permuted MGW tables (keeping reverse-complement symmetry) and found that none gave a performance increase as large as that of the true MGW table

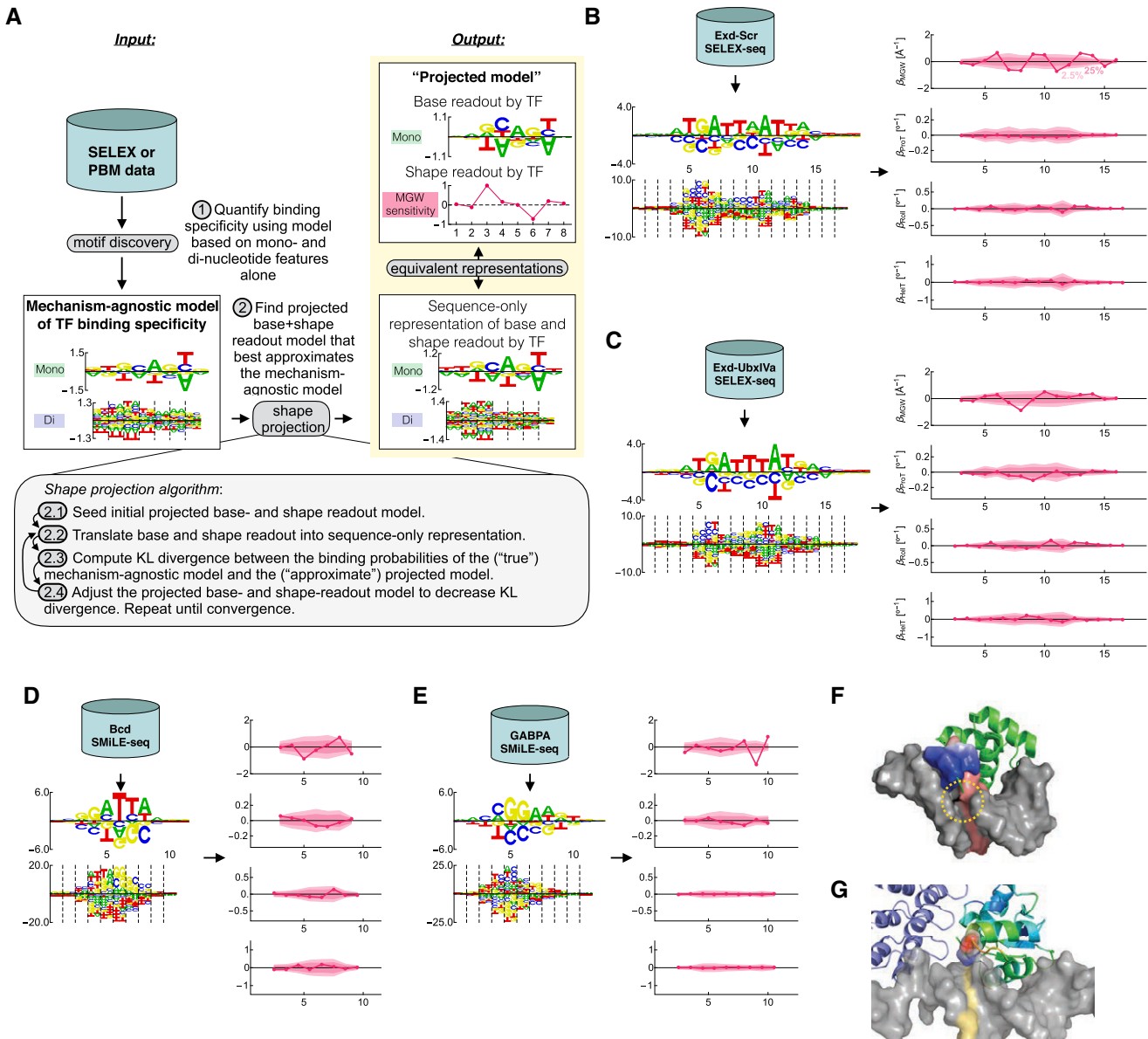

**Figure 5. Shape-sensitivity profiles inferred using shape projection.**

A   Schematic diagram showing the workflow for *post hoc* shape-readout analysis.

B   Mechanism-agnostic model for Exd-Scr (logos) and shape-sensitivity profiles ($\beta$) inferred using penalized shape projection in (A). Shaded areas indicate percentile levels (2.5, 25, 75, and 97.5%) of the distribution of shape-sensitivity profiles inferred using random sequence-to-shape models (see Materials and Methods and Fig EV4B).

C   The same as (B) but for Exd-UbxIVa.

D, E   The same as (B) but using SMiLE-seq data for the *Drosophila* homeodomain factor Bcd and the human ETS factor GABPα (Isakova *et al*, 2017).

F   NMR structure (1ZQ3, model 1) for Bcd showing an arginine residue (colored by electrostatic potential) inserted into the minor groove. The identified base pair 5 is highlighted in red. Two additional N-terminal arginine residues were not cloned for the structure but may extend along the minor groove (see dashed yellow circle).

G   Crystal structure (1AWC) for GABPα (right) in complex with GABPβ (left, blue). GABPα is colored by b-factor, which highlights a flexible loop (red–orange). The lysine residue, colored by electrostatic potential, is positively charged and inserted into the minor groove. The identified base pair at position 9 is highlighted in yellow.

(Fig 4E). Repeating the analysis for Exd-UbxIVa gave the same result (Fig EV2C).

Given the relatively simple structure of the shape tables revealed by our analysis (cf. Fig 3), however, we worried that random permutation of pentamers may not yield a proper null distribution. We reasoned that in order to stringently assess whether the shape

parameters play a privileged role as predictors of TF binding by a purely statistical criterion, they should be compared to randomized shape features of similarly lower complexity. Thus, we developed a procedure to generate random pentamer tables whose distribution of mono- and dinucleotide dependencies and the symmetry under reverse-complement transformation matched the real shape tables

(Materials and Methods and Fig EV4B). Repeating the unpenalized direct shape regression fits for Exd-Scr using these more sophisticated randomized tables outperformed the fit that used the true MGW table in 11% of the cases (Fig 4E), and therefore was unable to provide statistical significance for shape readout. Similarly, for Exd-UbxIVa 33% of random tables outperformed the real MGW table (Fig EV2C).

We reasoned that this lack of statistical significance could be due to poor conditioning of the unpenalized fits. Regularization tends to suppress overfitting and can help separating signal and background. Indeed, when we computed Exd-Scr shape-sensitivity profiles using $L_2$-regularized shape projection and repeated the analysis for five thousand random shape tables with matched complexity (Fig 5B), MGW readout in both the Exd and the Scr half-site was statistically significant even after multiple-testing correction across positions in the binding site (Benjamini–Hochberg FDR = 0.02 and 0.03, respectively). For Exd-UbxIVa, the readout was statistically significant in the Exd half-site but not in the UbxIVa half-site (FDR = 0.04; Fig 5C). Not all shape parameters showed statistically significant sensitivity; performing a similar analysis for propeller twist (ProT) revealed no significant sensitivity to variation in this parameter for either Exd-Scr or Exd-UbxIVa.

To more comprehensively assess our two-step approach of first building a mechanism-agnostic binding specificity model using *NRLB* followed by a statistical test of evidence of shape readout in this model using the *Shapelizer* tool introduced in this work, we first analyzed SELEX-seq data for MAX and found that no shape-sensitivity coefficients were significantly different from zero (Fig EV5A); the closest-to-significant coefficient was Roll at the central base pair step (FDR = 0.3), a position previously highlighted by Zhou *et al* (2015). We also analyzed HT-SELEX data for CEBPβ and found that the closest-to-significant coefficient was ProT at positions 5 and 10 (FDR = 0.12; Fig EV5B). Next, we analyzed SMiLE-seq binding data for the *Drosophila* transcription factor Bcd, and the human factors GABPα, GR, SP4, YY1, and ZEB1 (Isakova *et al*, 2017). Of these, GR, SP4, YY1, and ZEB1 did not have any shape-readout coefficients with an FDR below 0.1 (Fig EV5C–F).

For Bcd, the strongest preference for a narrowed minor groove was located at the underlined position in GG*A*TTA (FDR = 0.087; Fig 5D). Intriguingly, inspection of a structure for the Bcd–DNA complex solved using NMR spectroscopy (Baird-Titus *et al*, 2006) revealed that the N-terminal arm of Bcd is unstructured and makes contacts with the minor groove close to the base pair identified by *Shapelizer* (Fig 5F). In addition, the full-length protein used in the SMiLE-seq experiment had two additional arginine residues in the N-terminal domain that are in a suitable position to explain the identified MGW readout, but which were not cloned for the NMR structure.

For GABPα, the preference for a minor narrow groove was located at the underlined position in CCGGAA*G* (FDR = 0.002; Fig 5E). Inspection of a crystal structure of the GABPα–GABPβ heterodimer in complex with DNA (Batchelor *et al*, 1998) revealed that a leucine residue in the loop between α-helices 1 and 2 is interacting with the minor groove (Fig 5G). This again provides a possible structural rationale for the MGW readout inferred by *Shapelizer*.

Taken together, these results illustrate how penalized shape projection combined with a complexity-matched null model allows for robust identification of MGW readout, and how accounting for the precise nature of the sequence-to-shape relationship is necessary to assess the statistical significance of putative shape readout.

## Discussion

In this study, we revisited the problem of identifying DNA shape readout using TF binding data. Our insight that scoring matrices implicitly capture associations between DNA shape and binding affinity may not be necessarily perceived as surprising: A scoring matrix by definition summarizes the binding affinity of the highest-affinity sequence and the effect of single-base substitutions; since each of these sequences has a well-defined average shape, it follows naturally that the scoring matrix captures the difference in shape between the high-affinity sequences and random sequences, which have uniform average shape profiles. Nevertheless, we believe this insight to be important: Our analyses clearly show that associations between shape and binding affinity do not complement the scoring matrix, but rather are encodable in terms of mononucleotide features alone to a large extent (cf. Fig 3). This implies that model-based quantitative analysis of shape readout can be extended to transcription factors for which only a traditional scoring matrix is known (cf. Fig 2). To the best of our knowledge, this has not been shown explicitly before.

The feasibility of identifying shape readout using binding data alone is largely determined by how distinctive the signatures of shape and base readout are. Consider two extreme cases: In the first, DNA shape readout is fully captured by a mononucleotide sequence-to-shape model with no dependencies. Distinguishing shape and base readout will then be impossible without further context, such as structural information or binding data for mutated TFs. In the second extreme case, highly specific higher-order sequence features such as individual pentamers might determine DNA shape. Shape readout could then be attributed with high statistical confidence, since the set of equivalent higher-order features is large. In the real world, we are somewhere between these two extremes; our observation that most of the variation of DNA shape with base sequence can be explained using linear models based on mononucleotide features, analogous to scoring matrices, and that almost all residual variance can be explained by including interactions between neighboring nucleotide positions puts an upper limit on our ability to distinguish between base and shape readout based on DNA binding specificity alone. Depending on the TF and shape feature under consideration, it may or may not be possible to come to a clear conclusion without additional information.

As a practical conclusion, we advocate based on our overall insights that TF binding specificity first be quantified using a mechanism-agnostic mono- and dinucleotide readout model and that shape-readout analysis be implemented as *post hoc* analysis step which we refer to as "shape projection". In the cases of Exd-Scr and Exd-UbxIVa, this method identified previously known readout of minor groove narrowing. An argument against including dinucleotide terms in TF binding models is that it increases the number of parameters to be estimated (Zhou *et al*, 2015). However, while there are four mononucleotides and 16 dinucleotides, the number of independent parameters required to parameterize mono- and dinucleotide substitution effects equals 3 and 9 per position,

respectively. The advantage of including shape readout in reducing the number of model parameters is therefore more modest than one might think.

That there are only nine independent dinucleotide parameters is also important for understanding how much statistical power one has to distinguish between shape readout and incidental similarities of the TF binding specificity and the sequence dependences of the shape tables. In the case of Exd-Scr, we found that the performance improvement in the unpenalized binding model with MGW readout was not significantly larger than when matched random sequence features were used instead. In contrast, our penalized shape projection procedure identified statistically significant readout at the expected positions. What explains this apparent discrepancy? The lack of significance of the unpenalized shape regression can be understood by noting, first, that most shape readout is encoded in the scoring matrix; second, that this mononucleotide shape-readout component is collinear with base readout; and, finally, that the performance boost in the unpenalized model therefore is determined by the smaller dinucleotide shape-readout component and the degree to which it stands out to other unrelated dinucleotide interactions. In the unpenalized model, the shape-sensitivity coefficients can freely take any value and the nine independent dinucleotide interactions can easily be overinterpreted. By introducing penalization, the inferred shape sensitivity has to both explain the observed dinucleotide interactions and compete with the base readout to explain the mononucleotide preferences in the mechanism-agnostic model, all while keeping the parameter values as small as possible. The penalized shape projection therefore focuses on the most robust signatures of shape readout, and this improves significance.

In a recent study, Yang *et al* (2017) introduced "DNA shape logos" as a visualization of DNA shape readout. These logos show the improvement in model performance—as measured in terms of the improvement in $R^2$—that is observed when individual DNA shape features are introduced as predictors along the TF-DNA interface. The shape-sensitivity profiles instead show the change in binding free energy $\Delta\Delta G/RT$ per unit change in shape parameter. One practical difference between these measures is that the sign of the shape-sensitivity coefficient indicates whether a deviation in DNA shape is stabilizing or destabilizing, whereas the shape logo $\Delta R^2$, which always is positive, does not. A second difference is that the shape-sensitivity coefficient is a dimensionful quantity and therefore links the magnitude of a deviation in DNA shape to the resulting change in binding affinity. Finally, at least in our own (unpenalized) implementation of direct shape regression (cf. Fig 4), the shape coefficient values themselves occasionally showed unstable behavior due to confounded sequence and shape features (cf. Fig EV3). Our shape projection methodology (cf. Fig 5) addresses this problem and makes it possible to report stable shape coefficient values, along with measures of statistical significance. This is a useful improvement over existing methods.

While this manuscript was in review, two studies generated pentamer lookup tables for additional DNA shape parameters and for the electrostatic potential in the minor groove (Chiu *et al*, 2017; Li *et al*, 2017). It should be straightforward to use our *Shapelizer* software to analyze readout of these additional structural features.

In summary, studies integrating structural information with binding data for mutated proteins have in recent years unambiguously demonstrated that shape readout is an important sequence-readout mechanism. We here systematically examined the sequence dependences of the DNA shape parameters, how these dependences relate to signatures of shape readout, and how base and shape readout can be distinguished using TF binding data. We hope that our insights and methods will help expand knowledge of DNA shape readout.

## Materials and Methods

### SELEX binding-model inference using NRLB

*No Read Left Behind* (*NRLB*) is a software package that builds biophysical models of TF binding specificity directly from sequencing data generated using modern SELEX methods (SELEX-seq, HT-SELEX, and SMiLE-seq; Isakova *et al*, 2017; Jolma *et al*, 2010; Slattery *et al*, 2011; Zhao *et al*, 2009). A detailed description of *NRLB*, along with benchmarking results and biological applications of such models, is being prepared for separate publication.

The SELEX process starts with a library of DNA probes containing a *L*-bp random region flanked by constant adaptor sequences (Jolma *et al*, 2010; Slattery *et al*, 2011). This library is incubated with the TF of interest, and bound DNA probes are isolated. The bound probes (called round one) and a set-aside fraction of the initial library (called round zero) are then sequenced. *NRLB* infers sequence-to-affinity models using the sequenced round-zero and round-one probe counts. It does so by assuming (i) that the selection $\kappa_i$ of probe $i$ in round one is proportional to the total binding affinity of the probe (see below), and (ii) that the observed counts in both rounds follow a multinomial distribution:

$$\mathcal{L}(\text{round } r \text{ data}) = \prod_i p_{i,r}^{c_{i,r}}.$$

Here, $r = 0,1$ denotes the SELEX round, and the index $i$ runs over all $4^L$ probes; for each probe $i$, $c_{i,r}$ is the probe count. The probe frequency $p_{i,r}$ in round zero and round one, respectively, are related by

$$p_{i,1} = \frac{p_{i,0}\kappa_i}{Z_1},$$

where $Z_1 = \sum_i p_{i,0}\kappa_i$ is a normalizing partition function. The goal of *NRLB* is to learn a sequence-to-affinity model that optimally predicts probe selection $\kappa_i$.

In an ideal experiment, the probability of observing probes in the initial round $p_{i,0}$ would be constant across probes. However, actual experiments have significant round-zero bias. To correct for this, *NRLB* learns a bias model of the form

$$p_{i,0} = \frac{w_i}{Z_0},$$

$$w_i = \exp\left[\sum_{\varphi \in \Phi_0} \beta_\varphi^{(0)} X_{i\varphi}\right],$$

where $Z_0 = \sum_i w_i$. Here, $\varphi$ runs over the set $\Phi_0$ of all hexamers, $X_{i\varphi}$ represents the number of times each hexamer $\varphi$ occurs in probe $i$,

and $\beta_\varphi^{(0)}$ is a bias model parameter to be estimated from the round-zero data. *NRLB* first estimates $\beta_\varphi^{(0)}$ by maximizing the likelihood of the round-zero sequencing data defined above. For the HT-SELEX data, the bias model used tetramers as predictors (instead of hexamers) as this described the round-zero data better. For the SMiLE-seq data, the bias model used dimers as predictors since the variable region is 30 bp long and total probe length was restricted to 32 bp in the version of NRLB that we used for this study.

Next, we need to specify how the probe selection $\kappa_i$ depends on the probe sequence. Probe selection is modeled in terms of binding free energy by additively considering non-specific and specific binding contributions over all possible offsets and orientations $v$ within the probe:

$$\kappa_i = e^{\beta_{NS}^{(1)}} + \sum_v \exp\left[\sum_{\varphi \in \Phi_1} \beta_\varphi^{(1)} X_{iv\varphi}\right].$$

Here, $e^{\beta_{NS}^{(1)}}$ represents non-specific binding. For each binding view $v$, the free energy of binding (i.e., the exponent) is a linear function of the mono- and dinucleotide and shape features $\varphi$ evaluated in a window of width $k$. Thus, each term in sum over $\varphi$ is a product of predictor $X_{iv\varphi}$, which is a binary indicator for mono- or dinucleotide features and a continuous value for shape features; a binding-model parameter $\beta_\varphi^{(1)}$ corresponds to the free-energy contributions of each feature. *NRLB* estimates these binding-model parameters by maximizing the round-one likelihood using its gradient and dynamic programming methods. In addition, *NRLB* can impose reverse-complement symmetry on the binding model when analyzing SELEX data for homodimers such as MAX. In this study, we model binding affinity with three different sets of features $\Phi_1$: mononucleotide sequence; mononucleotide and dinucleotide sequence; and mononucleotide sequence and DNA shape features.

Note that while earlier methods have modeled TF binding using non-specific binding and summation over binding modes (Foat *et al*, 2006; Zhao *et al*, 2009; Riley *et al*, 2015; Ruan *et al*, 2017) and others have inferred TF binding models from SELEX data (Jolma *et al*, 2010; Slattery *et al*, 2011; Alipanahi *et al*, 2015; Zhou *et al*, 2015), *NRLB* has two distinct advantages: Firstly, it models the full set of SELEX probes without reducing the data to $k$-mer count tables (which makes analysis of wide binding sites infeasible since the number of $k$-mers increases exponentially with $k$), and secondly, it can include shape parameters in the free-energy model. Also note that while methods utilizing $k$-mer enumeration benefit from performing multiple rounds of SELEX selection, NRLB can infer reliable binding models after a single SELEX round, thus increasing its sensitivity to low-affinity binding.

### Metropolis–Hastings sampling of binding sequences

To investigate how scoring matrices encode associations between binding affinity and DNA shape, we wished to sample random sequences, sort them into free-energy bins, and compute the mean shape in each bin (cf. Fig 2B). Given a free-energy scoring matrix $w_{j,c}$ of width $k$, where $j = 1 \ldots k$ is the position within the binding site covered by the matrix and $c$ is the base at position $j$, we thus sampled sequences $s$ of length $k$ from the uniform distribution $p(s) = 1/4^k$ and binned them using the free-energy score $W(s) = \sum_j w_{j,s_j} \in [W_{\min}, 0]$, where $s_j$ is the base at position $j$ in $s$.

Because it is exceedingly rare to sample sequences in the extreme $W$ bins from the uniform distribution, we instead used the Metropolis–Hastings algorithm to sample sequences from the Boltzmann distribution $p(s; \beta_T) \propto \exp(\beta_T W(S))$ for different values of the inverse-temperature parameter $\beta_T$ (this populates all $W$ bins for positive and negative values of $\beta_T$) and then used rejection sampling (for each $W$ bin and $\beta_T$ value independently) to retain sequences following the uniform distribution within each bin. Altogether, 10 uniform $W$ bins and 13 uniformly spaced values $\beta_T \in \frac{\log 4^k}{-W_{\min}} * [-2, 2]$ were used. This sampling algorithm is implemented in the program sampleSequences.py.

### Shape of high-affinity SELEX-seq and UniPROBE probes

To identify high-affinity SELEX-seq probes, we computed the binding affinity of all 10mers using the SELEX package (Riley *et al*, 2014, 2015; http://bioconductor.org/packages/SELEX), retaining all 10mers with count 100 or greater and relative affinity 0.1 or greater. We then selected all full-length probes containing a high-affinity 10mer. To align the selected probes, we first aligned the high-affinity 10mers by maximizing the number of bases matching the top 10mer, then constructed a frequency matrix from these alignments, and, finally, used this frequency matrix to align the retained SELEX probes. For the UniPROBE and CIS-BP PBM datasets, we selected the 100 probes with the strongest binding signal and aligned these to the curated scoring matrix. We finally used the pentamer tables to compute the mean shape-parameter profile of the aligned SELEX and UniPROBE sequences. The method for computing the mean shape of sequences is implemented in the program kMerLinearRegression.py.

### Multivariate linear regression analysis of shape tables

Let $\varphi_a(s)$ denote the value of shape parameter $a$ for the pentamer sequence $s$. This was modeled using multiple linear regression models of the form $\hat{\varphi}_a(s) = \vec{X}(s) \cdot \vec{\gamma}_a$, where $\hat{\varphi}_a(s)$ is the predicted value of $\varphi_a(s)$, while $\vec{X}(s)$ represents row $s$ in the design matrix and $\vec{\beta}_a$ the parameters shown in Fig 3. The mononucleotide model has a design matrix $\vec{X}^{(0,1)}(s) = \{1, \delta_A^{s_1}, \ldots \delta_T^{s_5}\}$, where $\delta_n^m$ is the Kronecker delta function and $s_i$ is the base at position $i$. The design matrix for the dinucleotide model, $\vec{X}^{(0,1,2)}(s)$, was constructed by appending $\{\delta_{AA}^{s_{1:2}}, \ldots, \delta_{TT}^{s_{4:5}}\}$ to $\vec{X}^{(0,1)}(s)$, where $s_{i:i+1}$ is the dinucleotide starting at positon $i$. Finally, each row in the design matrix for the all-by-all model was constructed using the tensor product $\vec{X}^{(a)}(s) = \vec{X}^{(0,1)}(s) \otimes \vec{X}^{(0,1)}(s)$.

The model parameters were then computed using the formula $\gamma_a = (X^T \cdot X)^{-1} X^T \cdot \varphi_a$, where dot products indicate sums over pentamer and the inverse was computed by inverting all nonzero eigenvalues. For the step parameters Roll and HelT, the original shape tables report two values for each pentamer: the value at the step between the second and third base pair, and the value between the third and fourth. To get a reverse-complement symmetric tetramer table centered on the relevant base pair step, we computed the average of these two values after marginalizing over the right- and leftmost bases to align the step.

The zero eigenvalues in $(X^T \cdot X)$ are due to redundancies in the parametrization. The number of independent parameters is $\lceil (3k + 1)/2 \rceil$ in the reverse-complement symmetric mononucleotide model and $4\lfloor 3k/2 \rfloor - 2$ in the dinucleotide model. This can be

checked by counting the nonzero eigenvalues in $(X^T \cdot P_{RC} \cdot X)$, where $P_{RC}$ is a matrix that projects onto the space of symmetric k-mer tables. To control for the possibility of overfitting, we used an exhaustive leave-one-out cross-validation scheme where each pair of sequences related through reverse-complement transformation (which have the same value in the table) were left out separately. Linear regression on k-mer tables is implemented in the program kMerLinearRegression.py. The shape tables were generated by querying the DNAshape webserver for all pentamer sequences (Zhou *et al*, 2013).

## Quantifying binding-model performance

While the log-likelihood of the NRLB model in principle measures how well a binding model fits the SELEX-seq data, this value is difficult to interpret since even the true model cannot predict the sequencing shot noise and therefore has low log-likelihood. We instead quantified the "true" TF binding by sliding a 10-bp window across all SELEX probes, tabulating the counts, and retaining 10mers with count 100 or higher (for which the shot noise is small). While the 10mer table can be difficult to interpret since a strong binding site contributes to multiple shifted 10mers, note that a 10mer table can serve as a "fingerprint" for arbitrary sequence-to-affinity relationships (assuming nucleotide positions spaced 10 bp or more apart do not interact) and that accurate prediction of it indicates good model performance. We therefore quantified model performance by first tabulating the expected 10mer counts by sliding a 10-bp window over all possible probes weighted by the NRLB probe probability $a\left(s_i; \vec{\beta}\right) p_0(s_i)$, and then computing the Pearson correlation $r$ between the observed and modeled counts. To control for overfitting, half of the reads from SELEX round one were held out from the NRLB fitting and used for this calculation. To test whether differences in $r$ between models were significant, we applied the Fisher $r$-to-$z$ transformation to the $r$-values and then used a two-tailed $z$-test for significance.

## Shape projection

Given a mechanism-agnostic mono- and dinucleotide binding model, shape projection is a post-processing step that identifies the mononucleotide-plus-shape model that best approximates the mechanism-agnostic model. For the latter, the free energy of binding to a sequence $s$ is given by (cf. "*Mechanism-agnostic model of TF binding specificity*" in the lower left box in Fig 5A),

$$-\frac{\Delta\Delta G^{\mathrm{agn}}(s)}{RT} = \beta^{\mathrm{agn},(0)} + \sum_i \beta^{\mathrm{agn},(1)}_{i,s_i} + \sum_i \beta^{\mathrm{agn},(2)}_{i,s_{i:i+1}} \equiv \vec{X}^{(0,1,2)}(s) \cdot \vec{\beta}^{\mathrm{agn}},$$

where $\beta^{\mathrm{agn},(1)}_{i,s_i}$ is the scoring matrix and $\beta^{\mathrm{agn},(2)}_{i,s_{i:i+1}}$ parameterizes the dinucleotide interactions; the vector representation $\vec{\beta}^{\mathrm{agn}} = \left\{ \beta^{\mathrm{agn},(0)}, \vec{\beta}^{\mathrm{agn},(1)}, \vec{\beta}^{\mathrm{agn},(2)} \right\}$ was used on the right-hand side. The *projected* model (cf. "*Projected model*" in the top right box in Fig 5A) has a free energy of binding given by

$$-\frac{\Delta\Delta G^{\mathrm{proj}}(s)}{RT} = \beta^{\mathrm{proj},(0)} + \sum_i \beta^{\mathrm{proj},(1)}_{i,s_i} + \sum_i \beta^{\mathrm{proj},(\varphi)}_i \varphi(s_{i-2:i+2}),$$

where $\beta^{\mathrm{proj},(\varphi)}_i$ denotes the shape-sensitivity coefficient for feature $\varphi$ at position $i$ (step features, which use tetramers instead of

pentamers, are suppressed for brevity). It is convenient to formulate the projected model using mono- and dinucleotide variables; substituting in the mono- and dinucleotide sequence-to-shape model

$$\varphi(s_{i-2:i+2}) = \gamma^{(0)} + \sum_{\delta=-2}^{2} \gamma^{(1)}_{\delta,s_{i+\delta}} + \sum_{\delta=-2}^{1} \gamma^{(2)}_{\delta,s_{i+\delta:i+\delta+1}},$$

and reordering the terms yields (cf. "*Sequence-only representation of shape and base readout by TF*" in the lower right box in Fig 5A)

$$-\frac{\Delta\Delta G^{\mathrm{proj}}(s)}{RT} = \beta^{\mathrm{seq},(0)} + \sum_i \beta^{\mathrm{seq},(1)}_{i,s_i} + \sum_i \beta^{\mathrm{seq},(2)}_{i,s_{i:i+1}} \equiv \vec{X}^{(0,1,2)}(s) \cdot \vec{\beta}^{\mathrm{seq}},$$

where

$$\beta^{\mathrm{seq},(0)} = \beta^{\mathrm{proj},(0)} + \sum_i \beta^{\mathrm{proj},(\varphi)}_i \gamma^{(0)},$$

$$\beta^{\mathrm{seq},(1)}_{i,c} = \beta^{\mathrm{proj},(1)}_{i,c} + \sum_{\delta=-2}^{2} \beta^{\mathrm{proj},(\varphi)}_{i-\delta} \gamma^{(1)}_{\delta,c},$$

$$\beta^{\mathrm{seq},(2)}_{i,c:d} = \sum_{\delta=-2}^{1} \beta^{\mathrm{proj},(\varphi)}_{i-\delta} \gamma^{(2)}_{\delta,c:d},$$

and $\vec{\beta}^{\mathrm{seq}} = \left\{ \beta^{\mathrm{seq},(0)}, \beta^{\mathrm{seq},(1)}_i, \beta^{\mathrm{seq},(2)}_i \right\}$. Because shape readout at the first and last two positions of the binding model gives rise to mono- and dinucleotide specificity beyond of the mechanism-agnostic binding model, these readout coefficients were put to zero.

We next defined a loss function $V\left(\vec{\beta}^{\mathrm{seq}}; \vec{\beta}^{\mathrm{agn}}\right)$ to quantify how well the projected model $\vec{\beta}^{\mathrm{seq}}$ approximates the mechanism-agnostic model $\vec{\beta}^{\mathrm{agn}}$. Two choices of $V$ were considered:

(i)  *Affinity projection*: The projected model is defined to minimize the squared affinity error across all probes:

$$V_{\mathrm{affinity}}\left(\vec{\beta}^{\mathrm{seq}}; \vec{\beta}^{\mathrm{agn}}\right) = \sum_s \left( e^{\vec{X}^{(0,1,2)}(s) \cdot \vec{\beta}^{\mathrm{agn}}} - e^{\vec{X}^{(0,1,2)}(s) \cdot \vec{\beta}^{\mathrm{seq}}} \right)^2$$

$$= \sum_s \left( e^{2\vec{X}^{(0,1,2)}(s) \cdot \vec{\beta}^{\mathrm{agn}}} + e^{2\vec{X}^{(0,1,2)}(s) \cdot \vec{\beta}^{\mathrm{seq}}} - 2 e^{\vec{X}^{(0,1,2)}(s) \cdot \left(\vec{\beta}^{\mathrm{agn}} + \vec{\beta}^{\mathrm{seq}}\right)} \right) \cdot$$

$$= \sigma\left(2\vec{\beta}^{\mathrm{agn}}\right) + \sigma\left(2\vec{\beta}^{\mathrm{seq}}\right) - 2\sigma\left(\vec{\beta}^{\mathrm{agn}} + \vec{\beta}^{\mathrm{seq}}\right)$$

Here, the sums $\sigma\left(\vec{\beta}\right) \equiv \sum e^{\vec{X}^{(0,1,2)}(s) \cdot \vec{\beta}}$, which each contain $4^k$ terms, can be evaluated in $O(k)$ steps using dynamic programming. The gradient of the sums, $\vec{\nabla}\sigma\left(\vec{\beta}\right)$, is also straightforward to evaluate using dynamic programming.

(ii) *KL projection*: The Kullback–Leibler divergence $D_{\mathrm{KL}}(P||Q) = \sum_i P_i \log P_i/Q_i$ measures how much information is lost when the probability distribution $Q$ is used to approximate the true distribution $P$. In the context of multinomial regression, the KL divergence corresponds to the change in the expected model likelihood when the true probabilities $P$ are approximated with $Q$. To compare the mechanism-agnostic model and the projected model, we therefore consider the probability of drawing sequence $s$ assuming the sequences are sampled according to the binding affinities predicted in model $\vec{\beta}$:

                                                                    

$$p\left(s;\vec{\beta}\right) = \frac{e^{\vec{X}^{(0,1,2)}(s)\cdot\vec{\beta}}}{\sum_r e^{\vec{X}^{(0,1,2)}(r)\cdot\vec{\beta}}}.$$

We define the loss function to be the DL divergence between the mechanism-agnostic and projected sampling probabilities:

$$V_{KL}\left(\vec{\beta}^{\text{seq}};\vec{\beta}^{\text{agn}}\right) = \sum_s p\left(s;\vec{\beta}^{\text{agn}}\right)\ln\frac{p\left(s;\vec{\beta}^{\text{agn}}\right)}{p\left(s;\vec{\beta}^{\text{seq}}\right)}$$

$$= \left(\vec{\beta}^{\text{agn}} - \vec{\beta}^{\text{seq}}\right)\cdot\frac{\vec{\nabla}\sigma\left(\vec{\beta}^{\text{agn}}\right)}{\sigma\left(\vec{\beta}^{\text{agn}}\right)} - \ln\frac{\sigma\left(\vec{\beta}^{\text{agn}}\right)}{\sigma\left(\vec{\beta}^{\text{seq}}\right)}.$$

### Penalized regression

$L_1$ and $L_2$ regularization of a model with parameter vector $\vec{\beta}$ is accomplished by adding a penalty term $\lambda\parallel\vec{\beta}\parallel_p$ to the loss function $V\left(\vec{\beta}\right)$. We here consider two distinct types of parameter, namely the (mononucleotide) scoring matrix $\beta_{i,c}^{\text{proj},(1)}$ and the shape-sensitivity profile $\beta_i^{\text{proj},(\varphi)}$. We therefore computed the *p*-norm and penalized these parameters separately:

$$\parallel\beta^{\text{proj},(1)}\parallel_p = \sum_{i,c}\left|\beta_{i,c}^{\text{proj},(1)}\right|^p,$$

$$\parallel\beta^{\text{proj},(\varphi)}\parallel_p = \sum_i\left|\beta_i^{\text{proj},(\varphi)}\right|^p.$$

Scale parameters $\lambda$ determine the importance of these penalty terms relative to the loss function. Because these terms can have different units, it is convenient to normalize them to the same reference scale. As a reference, we used the binding model inferred using the unpenalized projection, $\beta^{\text{proj},\lambda=0}$, and finally used the combined loss function:

$$\frac{V\left(\vec{\beta}^{\text{seq}};\vec{\beta}^{\text{agn}}\right)}{V\left(\vec{\beta}^{\text{seq},\lambda=0};\vec{\beta}^{\text{agn}}\right)} + \lambda_{\text{shape}}\frac{\parallel\beta^{\text{proj},(\varphi)}\parallel_p}{\parallel\beta^{\text{proj},(\varphi),\lambda=0}\parallel_p} + \lambda_{\text{mono}}\frac{\parallel\beta^{\text{proj},(1)}\parallel_p}{\parallel\beta^{\text{proj},(1),\lambda=0}\parallel_p/4}.$$

By default, we set $\lambda_{\text{shape}} = \lambda_{\text{mono}} = 1$, but we also consider other values (cf. Fig EV6). The penalty term is not smooth in the case $p = 1$, and we instead solved the dual problem. For minimization, we used the SLSQP method from the SciPy library in Python. The penalized shape projection method is implemented as shapeProjection.py.

### Randomized k-mer tables

To generate random k-mer tables where the sequence-to-number relationship is of similar complexity as that observed in the true shape tables, recall that a linear mono- and dinucleotide model $\hat{\varphi}(s) = \vec{X}(s)\cdot\vec{\gamma}$ described the shape parameter $\varphi(s)$ well. We thus generated random k-mer tables by drawing parameter vectors $\vec{\gamma}$ from a random distribution. We initially drew vector components from the standard uniform distribution. However, this does not preserve important structures in the shape tables; it does not preserve the reverse-complement symmetry, it does not preserve

strong localization of the dependence at the central base or base pair step, and it does not preserve the relative strength of mono- and dinucleotide dependencies. To solve the first issue, we simply reverse-complement-symmetrized the table. To solve the second two issues, we quantified how the sequence dependence is distributed along the k-mer (see below) and then scaled the $\vec{\gamma}$ component, so the distributions matched between the random and true tables.

To characterize (and match) the sequence-to-shape relationship in a table $\varphi$, recall that the coefficient of determination, defined as the proportion of variance explained by a model, is a measure of model performance. Inspired by this, we asked: How much of the variance in shape-parameter values $\text{Var}_s[\varphi(s)]$ remains after conditioning on the base identity $c$ at position $i$ (i.e., how large is $\text{Var}_s[\varphi(s)|s_i = c]$)? How much remains if two bases are conditioned upon? The conditioned variance should be much smaller than the full variance if the conditioned-upon bases are important, and unchanged if the base is unimportant. We therefore used the expected conditional variance

$$C_{i,j}(\varphi) \equiv \text{E}_{c,d}[\text{Var}_s[\varphi(s)|s_i = c, s_i = d]],$$

as a measure of how strongly k-mer table $\varphi$ depends on the bases at positions $i$ and $j$. To match the complexity of the sequence-to-shape relationship between the true table $\varphi(s)$ and the modeled table $\hat{\varphi}(s)$, we finally scaled the components in the parameter vector $\vec{\gamma}$ to minimize the difference between the expected conditional variances

$$D = \sum_{i,j}\left|C_{i,j}(\varphi) - C_{i,j}(\hat{\varphi}(\vec{\gamma}))\right|^2.$$

Specifically, the mono- and dinucleotide components of $\vec{\gamma}$ were scaled according to $\gamma_{i,c}^{(1)} \to \alpha_i^{(1)}\gamma_{i,c}^{(1)}$ and $\gamma_{i,c;d}^{(2)} \to \alpha_i^{(2)}\gamma_{i,c;d}^{(2)}$, and the scaling parameters $\alpha_i^{(1)}$ and $\alpha_i^{(2)}$ were optimized to minimize the difference $D$ under the constraint of reverse-complement symmetry. The method for generating matched random k-mer tables, as well as the method for permuting k-mer tables, is implemented in the python script randomKmerTable.py available at https://github.com/BussemakerLab/Shapelizer/.

### Data availability

The analyses in this study were based on publicly available DNA shape tables (Zhou *et al*, 2013); SELEX-seq data for Exd-Scr and Exd-UbxIVa (Slattery *et al*, 2011), as well as MAX (Zhou *et al*, 2015); HT-SELEX for CEBPβ (ID: CEBPB_ESW_TCAACC20NCAA; Yang *et al*, 2017); SMiLE-seq data for Bcd, GABPα, GR, SP4, YY1, and ZEB1 (Isakova *et al*, 2017); PBM data and a scoring matrix for HNF4A from UniPROBE (Robasky & Bulyk, 2011); and PBM data and a scoring matrix for Ftz-F1 from CIS-BP (Weirauch *et al*, 2014).

The code is available at http://github.com/BussemakerLab/Shapelizer.

**Expanded View** for this article is available online.

### Acknowledgements

We are grateful for valuable feedback on the manuscript from Remo Rohs and Richard Mann, as well as discussions with members of the Rohs, Mann, and

Bussemaker laboratories, and Xiang-Jun Lu and Gabriella Martini in particular. This research was supported by National Institutes of Health grant R01HG003008 to H.J.B. and an HHMI International Fellowship to J.F.K. Columbia University's Shared Research Computing Facility is supported by NIH grant G20RR030893 and NYSTAR contract C090171.

## Author contributions

HTR and HJB conceived and developed the analysis, with contributions from CR and JFK HTR and CR carried out the analysis of the SELEX-seq data. HTR and HJB wrote the manuscript with input from all authors.

## Conflict of interest

The authors declare that they have no conflict of interest.

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
