## [Review Process File · Molecular Systems Biology]

A unified approach for quantifying and interpreting DNA shape readout by transcription factors

H. Tomas Rube, Chaitanya Rastogi, Judith F. Kribelbauer and Harmen J. Bussemaker

Review timeline:

Submission date:	28 July 2017
Editorial Decision:	21 December 2017
Revision received:	7 November 2017
Editorial Decision:	15 December 2017
Revision received:	26 January 2018
Accepted:	31 January 2018

Editor: Maria Polychronidou

Transaction Report:

1st Editorial Decision

21 December 2017

Thank you again for submitting your work to Molecular Systems Biology. We have now heard back from the two referees who agreed to evaluate your study. As you will see below, the reviewers acknowledge that the study seems interesting. They raise however a series of concerns, which we would ask you to address in a major revision of the manuscript.

The reviewers' recommendations are rather clear so I think that there is no need to repeat all the points listed below. Some of the more fundamental issues raised are the following:

- Both reviewers suggest including additional analyses for some more TFs in order to demonstrate more convincingly that the conclusions are generalisable and that the approach can reveal new insights. Such analyses would definitely enhance the impact of the study.
- A particularly important point refers to the need to provide the code required for reproducing the results described in the manuscript.
- Both reviewers thought that the methodology is not described in enough detail. We would therefore ask you to make sure that all methodology that was used in the context of this work is clearly described in the Materials and Methods. Related to this point, if the study describing the "No Read Left Behind" method has been published in the meantime we would ask you to include the reference. If not, it should be cited in the text as Rastogi et al, in preparation. In any case, the NRLB methodology as used in this study should be clearly described so that it is accessible to the readers.

All other issues raised by the reviewers would need to be convincingly addressed. As you might already know, our editorial policy allows in principle a single round of major revision so it is essential to provide responses to the reviewers' comments that are as complete as possible.

REVIEWER REPORTS

Reviewer #1:

The authors present new methods and useful analyses for deconvolving the contribution of DNA base read out and DNA shape read out in the context of modeling in-vitro binding affinity of proteins to DNA. First, they show that mono and dinucleotide linear models can fit DNA shape parameters with high accuracy. These models provide compressed, interpretable sequence-based representations of the more complex shape parameters. The results also indicate that general (beyond mononucleotide) sequence-based in-vitro TF binding models can implicitly encode shape features. The primary advantage of shape features is then interpretation. However, the confounding between sequence-based base readout and shape read out, which is also explained by sequence, makes it difficult to deconvolve the two. The authors hence devise a post-hoc approach that they call "shape projection" to achieve interpretable deconvolution of base and shape read out. The main idea is to first fit a "mechanism agnostic" mono+dinuc sequence model to counts of 10mers from SELEX or PBM data. This model is based on an unpublished method called NRLB which is basically a multinomial logistic regression model. Then, the authors formulate another parameterized linear model called the "shape projected model" that explicitly encodes shape sensitivity parameters and approximates the mechanism agnostic model. The shape projected model can be expressed in terms of mono and dinuc features derived from the sequence-to-shape linear models. The goal is now to learn the parameters of the shape projected model by minimizing the squared error or the KL divergence between the mechanism agnostic model (true model) and the shape projected model (the approximation). To avoid colinearity and confounding effects, the authors use appropriately normalized L2 regularization. They use a null model based on randomized pentamer-shape tables to assess the statistical significance of the shape sensitivity parameters. The method is primarily evaluated on two gold standard datasets namely Exd-Scr and Exd-UbxIVa for which shape properties are well studied.

Overall, even though the paper is dense and covers some potentially confusing and subtle concepts, formulations and rather intricate and rigorous analyses, the paper is well written. The schematics in the figures are very helpful.

Below are a few comments/suggestions/required clarifications.

1. The authors use squared Pearson correlation (r^2) between observed and predicted 10mer counts to evaluate their NRLB models. I would like to see other measures of concordance that might be less sensitive to heteroscedasticity. The dynamic range of counts can skew r^2 to artificially high values. Can you compute the spearman correlation as well as the Pearson on log/asinh counts. Also visually displaying a few of the scatter plots of observed to predicted counts will be essential in the main figures or supplement.
2. Are the r^2 being reported on the training data or on held out data? This is not clear from the text. They should be evaluated on held out data. The authors do mention leave-one out cross validation for the sequence-to-pentamershape models but its not clear what they are using for the NRLB models in the paper.
3. Overall, one message of the paper is quite clear that shape is implicitly encoded in sequence based models of affinity. Hence, mechanism agnostic sequence models are appropriate and sufficient for quantifying binding affinity and specificity from in-vitro binding assays. The shape representation is clearly more relevant from an interpretation stand point than a predictive stand point.

Then, there is the more subtle deconvolution/contribution of shape read out in the TF affinity models using the KL divergence method. I can understand why the analysis in the paper focuses on just two main TF datasets since these are gold standards in terms of what is known about shape. However, it is a bit unsatisfying. This is not essential but I would encourage the authors to show more examples of how this modeling approach helps to interpret shape contributions for a few other TFs/datasets. Do the authors observe any novel interpretations that were not previously known?

4. The authors did not share a preprint of the unpublished NRLB paper to assist with the review. Moreover, no code is shared and there is no submission of the learned models or any supplementary website hosting these. This is primarily a methods paper on posthoc analysis of DNA shape read out models. So providing code that can be used to reproduce results and analyze new data is critically important. I would request the authors to provide this information in the revised version of the paper.

Reviewer #3:

Comments for the authors:

This manuscript presents an approach to compute the extent of shape-readout at transcription factor (TF) binding sites. It is a two-step approach that first develops a sequence-based model (mechanism-agnostic model) of TF binding affinity and then uses a sequence-to-shape model to quantify the extent of shape-readout from the mechanism-agnostic model. This is an interesting and timely question. The authors describe some novel approaches to the problem. Unfortunately, it is very challenging to reconstruct all the analyses, approaches and arguments from the manuscript in its current form.

Our major comments and concerns on the manuscript are as follows.

1. It is not clear how the end results on shape-readouts obtained by this post-hoc approach are different from and more interpretable than the shape-logos developed recently by Lin and Yaron et al. (PMID: 28167566). This question is relevant since the authors have used the same panels in Figs 4 and 5 to explain base and shape readout computation under the direct approach and the projection approach.

2. The manuscript needs extensive rewriting to explain key concepts and methodological steps. It was difficult in many places to reconstruct exactly what was proposed or done. For example:

- a. How should we interpret the shape-sensitivity coefficients?
- b. The authors used the term "scoring matrix" in some places to mean a mono-nucleotide model, and in other places to mean a mono + di nucleotide model. This is confusing and should be used with clear descriptions when used in statements like "Scoring matrices encode known associations between DNA shape and binding affinity".
- c. The Introduction should to be more focused and clearly describe the specific question being addressed.
- d. Expressions like "somewhat crudely" (in Pg. 17) sound informal and require clear elaboration.
- e. In pg 6, "A simple scoring matrix ... encodes a large part ... " -- how are the authors quantifying this?
- f. In pg 10, the part "In practical terms, this means that ... when underlying data is unavailable (such as when only a scoring matrix is known) ..." is not clear. If a scoring matrix is known, it must have originated from data and we do not know of a scenario when the data is not available but the matrix is.
- g. In pg 11, the part "In fact, convergence difficulties that we encountered ..." is not clear and comes completely out of the blue.
- h. In pg 13, "not all shape parameters showed statistically significant sensitivity" -- the authors should explain how they defined sensitivity here.
- i. Pg 16, in Hastings-Metropolis sampling, how was a value chosen for k?
- j. Pg 18, how was the part $p_0(s_i)$ obtained for the NRLB probe probability?

3. Results here are based on the NRLB model, which has not been published yet (and to the best of our knowledge, there is no preprint of NRLB). It is not clear why the authors chose to use NRLB here. The authors state that "NRLB is unique in that ... without the need for an intermediate oligomer enrichment table ..." (Pg 9), but this point of "intermediate oligomer enrichment table" is not clear. A clear explanation of the method or citation of another paper that includes this information is critical for readers to understand and assess the results presented. Also, it is not clear why NRLB considers SELEX probes after the first round of affinity selection -- why does NRLB not look into probes after more rounds of selection?

4. It is not clear why the authors chose to model only three datasets in the section "Scoring matrices encode known associations between DNA shape and binding affinity" (and why particularly those three datasets/TFs). A question remains about how generalizable these statements are across different TFs. Generally speaking, in this section the authors could have chosen those datasets where higher order models have been found in previous studies to yield the most gain in accuracy.

5. When the authors state "Scoring matrices encode known associations between DNA shape and binding affinity", they are actually talking about sites with strong sequence specificity. As the Methods section shows, these sites were selected based on enriched 10-mers. Unless the enriched 10-mers are very different, it is not surprising that the shape of these sites will be captured well by a mono-nucleotide based scoring matrix. These points should be checked and the claims corrected accordingly.

6. In the subsections "A mononucleotide model captures two-thirds of the variance in DNA shape parameters" and "Dinucleotide features predict shape parameters with high precision", for the table of a given shape feature, the authors are actually modeling ~500 data points (since these are 5-mer tables and we collapse each k-mer with its reverse complement) with >50 free parameters (considering both mono- and di-nucleotide parameters). Without getting a clear idea about the data, it is difficult to convince oneself that leave-one-out CV has taken care of generalization issues. Also, it is not clear why the authors did not try a systematic model selection approach using penalization for increasing complexity. While model selection should be tried (or explained why the authors did not go through these steps), the authors should also consider fitting equally complex models on randomly generated tables. The authors can also consider fitting their models on shape-profiles directly from the Rohs lab's MD simulations rather than on the averaged data in these 5-mer tables. That could give a better estimate on how much variance in shape-features can be accounted for by a straightforward sequence-to-shape models.

7. In several places (Figs 2 and 5, and related text) the authors made statements about shape-features in general, but showed results only on MGW or ProT. It is not clear why the authors did not show results from the other features.

8. The idea of shape projection is not clearly conveyed by Figure 5. The authors should consider providing a detailed visualization.

9. Providing code would help with reproducibility.

Typos:

Pg 2, paragraph 1: "and measuring in vitro affinities in high-throughput [3-7]" -> should be "... high-throughput assays ..."

Pg 5, paragraph 1: "minor grove"  "minor groove"

Pg 5, paragraph 2: "that complete neglects"  "that completely neglects"

Pg 13: "FRD"  FDR

Pg 16: "DNA shape shape profiles"  shape twice

Pg 18: "Given an mechanism agnostic"  "a mechanism agnostic"

Pg 19: "When probability the distribution Q  " "when the probability distribution Q"

Response to reviewer #1:

“The authors present new methods and useful analyses for deconvolving the contribution of DNA base read out and DNA shape read out in the context of modeling in-vitro binding affinity of proteins to DNA. First, they show that mono and dinucleotide linear models can fit DNA shape parameters with high accuracy. These models provide compressed, interpretable sequence-based representations of the more complex shape parameters. The results also indicate that general (beyond mononucleotide) sequence-based in-vitro TF binding models can implicitly encode shape features. The primary advantage of shape features is then interpretation. However, the confounding between sequence-based base readout and shape read out, which is also explained by sequence, makes it difficult to deconvolve the two. The authors hence devise a post-hoc approach that they call "shape projection" to achieve interpretable deconvolution of base and shape read out. The main idea is to first fit a "mechanism agnostic" mono+dinuc sequence model to counts of 10mers from SELEX or PBM data. This model is based on an unpublished method called NRLB which is basically a multinomial logistic regression model. Then, the authors formulate another parameterized linear model called the "shape projected model" that explicitly encodes shape sensitivity parameters and approximates the mechanism agnostic model. The shape projected model can be expressed in terms of mono and dinuc features derived from the sequence-to-shape linear models. The goal is now to learn the parameters of the shape projected model by minimizing the squared error or the KL divergence between the mechanism agnostic model (true model) and the shape projected model (the approximation). To avoid colinearity and confounding effects, the authors use appropriately normalized L2 regularization. They use a null model based on randomized pentamer-shape tables to assess the statistical significance of the shape sensitivity parameters. The method is primarily evaluated on two gold standard datasets namely Exd-Scr and Exd-UbxIVa for which shape properties are well studied.

Overall, even though the paper is dense and covers some potentially confusing and subtle concepts, formulations and rather intricate and rigorous analyses, the paper is well written. The schematics in the figures are very helpful “

We thank the reviewer for their careful reading of the manuscript.

“Below are a few comments/suggestions/required clarifications.

The authors use squared Pearson correlation (r^2) between observed and predicted 10mer counts to evaluate their NRLB models. I would like to see other measures of concordance that might be less sensitive to heteroscedasticity. The dynamic range of counts can skew r^2 to artificially high values. Can you compute the spearman correlation as well as the Pearson on log/asinh counts. Also visually displaying a few of the scatter plots of observed to predicted counts will be essential in the main figures or supplement.“

We agree that it is useful to compare different metrics. Following the reviewer's suggestion, the revised manuscript now reports squared Pearson correlation (r^2) both for the k-mer counts themselves and for the log-transformed k-mer counts (the original manuscript in fact showed the log-transformed counts, but the legend did not indicate this), as well as the Spearman correlation for the k-mer counts. These measures of correlation all give a very similar ordering of the models. We also repeated the analysis using asinh but decided to exclude the resulting figure as it was indistinguishable from the log-count figure (since the 10mer counts are large and $\text{asinh}(n) = \log(n) + \log(2)$ for $n \gg 1$). The revised manuscript also now contains scatter plots showing observed vs. predicted 10mer counts (Figure 4b).

"2. Are the r^2 being reported on the training data or on held out data? This is not clear from the text. They should be evaluated on held out data. The authors do mention leave-one out cross validation for the sequence-to-pentamershape models but its not clear what they are using for the NRLB models in the paper."

In the original manuscript we did not hold out data, because we did not expect any observable difference given the small number of parameters in the model. However, we agree with the reviewer that, strictly speaking, only metrics based on held-out data are meaningful. In the revised manuscript we therefore now hold out 50% of the reads in SELEX round one for the r^2 calculation. This did indeed not significantly change the coefficients of the binding models (the median absolute change in $\Delta\Delta G/RT$ was 0.0023). However, only using half the data for the r^2 calculation decreased the number of k-mers exceeding the 100-count threshold from 64,277 to 16,413 and r^2 values were shifted down slightly, although the relative order did not change. We have edited the legends for Figure 4 accordingly.

"3. Overall, one message of the paper is quite clear that shape is implicitly encoded in sequence based models of affinity. Hence, mechanism agnostic sequence models are appropriate and sufficient for quantifying binding affinity and specificity from in-vitro binding assays. The shape representation is clearly more relevant from an interpretation stand point than a predictive stand point."

Then, there is the more subtle deconvolution/contribution of shape read out in the TF affinity models using the KL divergence method. I can understand why the analysis in the paper focuses on just two main TF datasets since these are gold standards in terms of what is known about shape. However, it is a bit unsatisfying. This is not essential but I would encourage the authors to show more examples of how this modeling approach helps to interpret shape contributions for a few other TFs/datasets. Do the authors observe any novel interpretations that were not previously known?"

To demonstrate that *our shape projection* procedure is generalizable, we include additional analysis of Max (SELEX-seq) and CEBP β (HT-SELEX) in the revised manuscript.

"4. The authors did not share a preprint of the unpublished NRLB paper to assist with

the review. Moreover, no code is shared and there is no submission of the learned models or any supplementary website hosting these. This is primarily a methods paper on posthoc analysis of DNA shape read out models. So providing code that can be used to reproduce results and analyze new data is critically important. I would request the authors to provide this information in the revised version of the paper.

To address this concern we have created a python-based software package named *Shapelizer* that perform a number of tasks: (i) sampling and binning of random sequence using a scoring matrix (Figure 2); (ii) computation of the mean shape of sequences (Figure 2); (iii) linear regression of kmer tables (Figure 3); (iv) generation of random shape tables (Figure 3-5); and (v) shape projection (Figure 5). We have also included shell scripts and a Makefile that use the package to generate the data shown in Figures 2, 3, and 5 (making Figure 4 is computationally demanding and requires access to a computer cluster, whereas the other programs can be run on a standard laptop). The code is available at <https://github.com/BussemakerLab/Shapelizer/>.

To address the reviewer's concern regarding NRLB, we have expanded the methods section. We are also attaching a preprint of the NRLB manuscript in the resubmission for the reviewers to read. The NRLB manuscript is currently out for review. We plan to upload a preprint to *bioRxiv* once we have submitted a revised version in response to the initial reviews.

Response to reviewer #3:

“Comments for the authors:

This manuscript presents an approach to compute the extent of shape-readout at transcription factor (TF) binding sites. It is a two-step approach that first develops a sequence-based model (mechanism-agnostic model) of TF binding affinity and then uses a sequence-to-shape model to quantify the extent of shape-readout from the mechanism-agnostic model. This is an interesting and timely question. The authors describe some novel approaches to the problem. Unfortunately, it is very challenging to reconstruct all the analyses, approaches and arguments from the manuscript in its current form.”

We thank the reviewer for insightful comments and suggestions for improved presentation.

“Our major comments and concerns on the manuscript are as follows.

1. It is not clear how the end results on shape-readouts obtained by this post-hoc approach are different from and more interpretable than the shape-logos developed recently by Lin and Yaron et al. (PMID: 28167566). This question is relevant since the authors have used the same panels in Figs 4 and 5 to explain base and shape readout computation under the direct approach and the projection approach.”

The “shape logos” introduced by Yang, Orenstein, et al. (2017) show the improvement in model performance – as measured in terms of the improvement in R^2 – that is observed when individual DNA shape features are introduced as predictors along the TF-DNA interface. The shape sensitivity profiles instead show the change in binding free energy $\Delta\Delta G/RT$ per unit change in shape parameter. One practical difference between these measures is that the sign of the shape sensitivity coefficient indicates if a deviation in DNA shape is stabilizing or destabilizing whereas the shape-logo ΔR^2 , which always is positive, does not. A second difference is that the shape-sensitivity coefficient is a dimensionful quantity and therefore links the magnitude of a deviation in DNA shape to the resulting change in binding affinity. In the revised manuscript, we include a paragraph in the Discussion section that clarifies these differences.

Finally, in our own implementation of direct shape regression (Figure 4) and presumably also in the model-based analyses performed by other groups (including Yang, Orenstein et al., 2017), the shape coefficient values themselves can show unstable behavior due to confounding of sequence and shape features, which is a key motivation for reporting ΔR^2 values rather than coefficients. The methodology of Figure 5 addresses this problem, and makes it possible to report stable shape coefficient values, along with measures of statistical significance. This is a useful improvement over the existing methods.

“2. The manuscript needs extensive rewriting to explain key concepts and

methodological steps. It was difficult in many places to reconstruct exactly what was proposed or done. For example:

a. How should we interpret the shape-sensitivity coefficients?"

To clarify the interpretation of the shape-sensitivity coefficients we have added the sentence "[...], which have the interpretation of change in (normalized) binding free energy ($\Delta\Delta G/RT$) per unit (angular degree for Roll, HelT, ProT; Ångström for MGW) of change in shape-parameter value".

"b. The authors used the term "scoring matrix" in some places to mean a mono-nucleotide model, and in other places to mean a mono + di nucleotide model. This is confusing and should be used with clear descriptions when used in statements like "Scoring matrices encode known associations between DNA shape and binding affinity".

In the revised manuscript we define the term *scoring matrix* in the introduction: "we reserve the term *scoring matrix* for TF binding models that only have mononucleotide predictors". Figure 1 of the original manuscript could be misinterpreted to mean that a scoring matrix encodes both mono- and dinucleotide preferences. This has been fixed.

"c. The Introduction should to be more focused and clearly describe the specific question being addressed."

At the reviewer's suggestion, we have made a number of edits to the Introduction. We have also expanded the last paragraph of the Introduction to state the specific questions being addressed and to give a brief outline of the paper.

"d. Expressions like "somewhat crudely" (in Pg. 17) sound informal and require clear elaboration."

We have edited out this informal parenthetical remark.

"e. In pg 6, "A simple scoring matrix ... encodes a large part ... " -- how are the authors quantifying this?"

We have edited this sentence to be more qualitative: "[...] a simple scoring matrix somehow already encodes known differences in DNA shape between high- and low-affinity binding sequences."

"f. In pg 10, the part "In practical terms, this means that ... when underlying data is unavailable (such as when only a scoring matrix is known) ..." is not clear. If a scoring matrix is known, it must have originated from data and we do not know of a scenario when the data is not available but the matrix is."

One example of the situation we are referring to is that of online TF motif databases that curate scoring matrices but do not store the sequences upon which the matrix is based

(such as CIS-BP). Even if the data can be retrieved from other sources (by tracking down the original publication and hopefully getting access to the raw data), creating a pipeline to reprocess the raw data is in many cases impractical and can require significant expertise. However, we agree with the reviewer that the original statement “*underlying data is unavailable*” can be moderated and we edited the sentence in question to state: “*In practical terms, this means that [...] when underlying data is not readily available or impractical to reprocess (such as when online databases only curate the scoring matrix), [...].*”

“g. In pg 11, the part “*In fact, convergence difficulties that we encountered ...*” is not clear and comes completely out of the blue.”

We agree that this sentence comes out of the blue and it has been removed. The point is in fact important, because it makes the technical shortcomings of the direct shape regression approach in Figure 4 more explicit, and therefore provides motivation for the post-hoc analysis of Figure 5. We have therefore added the sentence “*Second, while including shape readout improved performance, the inferred sensitivity coefficients often appeared unstable (Figure EV3), suggesting that the problem was poorly conditioned*” to the paragraph motivating the shape projection method. We also discuss this point further in the Discussion.

“h. In pg 13, “*not all shape parameters showed statistically significant sensitivity*” -- the authors should explain how they defined sensitivity here.”

We have edited this paragraph to clarify that finding no “*statistically significant sensitivity*” means “*no shape-sensitivity coefficients was significantly different from zero*”.

“i. Pg 16, in Hastings-Metropolis sampling, how was a value chosen for k?”

k is the width of the scoring matrix and is not a free parameter given a scoring matrix. We have edited the relevant paragraph of the Methods section to state explicitly that k is the length of the scoring matrix.

“j. Pg 18, how was the part $p_{i,0}(s_i)$ obtained for the NRLB probe probability?”

The revised manuscript has an expanded description of the round-zero model. The round-zero probe counts $c_{i,0}$ are modeled using the multinomial distribution

$$\mathcal{L}(\text{round zero data}) = \prod_i p_{i,0}^{c_{i,0}}$$

where the probability $p_{i,0}$ is proportional to the concentration of probe i in round zero. In an ideal experiment $p_{i,0}$ would be constant across probes. However, real experiments have significant round-zero bias. To correct for this, NRLB learns a bias model of the form

$$p_{i,0} = \frac{w_i}{Z_0}$$

$$w_i = \exp \left[\sum_{\varphi \in \Phi_0} \beta_{\varphi}^{(0)} X_{i\varphi} \right]$$

and $Z_0 = \sum_i w_i$. Here φ runs over the set Φ_0 of all 6mers, $X_{i\varphi}$ is the number of times the 6mer φ occurs in probe i , and $\beta_{\varphi}^{(0)}$ is a bias parameter to be estimated. The first step in NRLB estimates $\beta_{\varphi}^{(0)}$ by maximizing the likelihood of the round-zero sequencing data defined above.

“3. Results here are based on the NRLB model, which has not been published yet (and to the best of our knowledge, there is no preprint of NRLB). It is not clear why the authors chose to use NRLB here. The authors state that “NRLB is unique in that ... without the need for an intermediate oligomer enrichment table ...” (Pg 9), but this point of “intermediate oligomer enrichment table” is not clear. A clear explanation of the method or citation of another paper that includes this information is critical for readers to understand and assess the results presented. Also, it is not clear why NRLB considers SELEX probes after the first round of affinity selection -- why does NRLB not look into probes after more rounds of selection?”

To address this concern, we have expanded the description of the NRLB algorithm in the methods section. The NRLB manuscript is currently out for review (for two months and counting). We plan to upload the preprint to bioRxiv once we have the reviews. For the purpose of reviewing this manuscript, we have attached a preprint of the NRLB manuscript in the revised submission.

One major advantage of NRLB compared to previous methods is it can infer a sequence+shape model directly from the sequencing data without constructing an intermediate oligomer enrichment table. This is especially important when analyzing TF complexes with extended footprints; for the Exd-Scr complex we infer a 18bp binding model. At that footprint width, tabulation of oligomers is not feasible as the counts for even the high-affinity sequences will get too low.

Finally, while later SELEX-rounds, are useful for constructing oligomer enrichment tables, we have found that only a single round of selection is required for NRLB to learn accurate TF binding models. Moreover, for technical reasons NRLB can only infer model from R1 data, and not from later rounds. This is not a limitation, as additional SELEX rounds actually tend to decrease sensitivity to low-affinity binding sequences. To clarify this, we have added the remark *“note that while methods utilizing k-mer count tables benefit from performing multiple rounds of SELEX-selection, NRLB can infer reliable binding models after a single SELEX round, thus increasing its sensitivity to low-affinity binding.”* to the Materials and Methods section.

“4. It is not clear why the authors chose to model only three datasets in the section “Scoring matrices encode known associations between DNA shape and binding affinity” (and why particularly those three datasets/TFs). A question remains about how generalizable these statements are across different TFs. Generally speaking, in this section the authors could have chosen those datasets where higher order models have

been found in previous studies to yield the most gain in accuracy.”

To address his concern, we have expanded the analysis in the relevant section to include a total of five TFs, namely Exd-Scr (SELEX-seq), Exd-UbxIVa (SELEX-seq), Max (SELEX-seq), Hnf4 (PBM from UniPROBE), and Ftz-f1 (PBM from CIS-BP). Together, these examples cover different TF families and several major experimental resources and platforms.

“5. When the authors state "Scoring matrices encode known associations between DNA shape and binding affinity", they are actually talking about sites with strong sequence specificity. As the Methods section shows, these sites were selected based on enriched 10-mers. Unless the enriched 10-mers are very different, it is not surprising that the shape of these sites will be captured well by a mono-nucleotide based scoring matrix. These points should be checked and the claims corrected accordingly.”

When we state *"Scoring matrices encode known associations between DNA shape and binding affinity"*, we refer to trends in DNA shape when comparing high affinity sequences to lower affinity sequences. The reviewer is correct that the binding probes displayed in Figure EV1 (Old Figure 2a) were selected because they contained a high-enrichment 10mer. The reviewer is also correct that it may not be surprising to some researches (at least in retrospect) that high-affinity SELEX-probes are similar to high-affinity sequences identified using a scoring matrix. Indeed, this is the reason why we choose to make the very same point in the first paragraph of the Discussion section. While this point may not be surprising to some, it was surprising to us. Moreover, we have found (discussing this issue at conferences) that most researchers seem to think that DNA shape encode something completely different than what is encoded in scoring matrices. We therefore think it is important to explicitly demonstrate that shape-trends are encoded in scoring matrices and to give a simple explanation of this in the discussion section.

“6. In the subsections "A mononucleotide model captures two-thirds of the variance in DNA shape parameters" and "Dinucleotide features predict shape parameters with high precision", for the table of a given shape feature, the authors are actually modeling ~500 data points (since these are 5-mer tables and we collapse each k-mer with its reverse complement) with >50 free parameters (considering both mono- and di-nucleotide parameters). Without getting a clear idea about the data, it is difficult to convince oneself that leave-one-out CV has taken care of generalization issues. Also, it is not clear why the authors did not try a systematic model selection approach using penalization for increasing complexity. While model selection should be tried (or explained why the authors did not go through these steps), the authors should also consider fitting equally complex models on randomly generated tables. The authors can also consider fitting their models on shape-profiles directly from the Rohs lab's MD simulations rather than on the averaged data in these 5-mer tables. That could give a better estimate on how much variance in shape-features can be accounted for by a straightforward sequence-to-shape models.”

A careful analysis of the number of independent parameters shows that the number of parameters in the mononucleotide model is 7 and 8 for the base-step and base parameters, respectively. Adding dinucleotide interactions increases these numbers to 22 and 26, respectively (after removing redundant parameters and imposing reverse-complement symmetry on the model the number of parameters is $6k-2$ and $6k-4$ for even and odd k , respectively). There are therefore ~ 20 data points per independent parameter for the base models and ~ 12 data points per parameter for the step models. This is, in our opinion, sufficient to get a good estimate of the model parameters. However, the reviewer does correctly highlight that the manuscript should be more clear about the number of independent parameters. To address this valid concern, we have added the number of independent parameters to the main text and the formulae to the methods sections.

Regarding the efficacy of the cross validation, it is not clear to us why leave-one-out cross validation (leaving out reverse-complement pairs) would not fully control for possibility of overfitting. However, the reviewer makes the prudent suggestion that we investigate the if models can find patterns in permuted data tables. We performed this calculation as the updated manuscript describes: *“Repeating the analysis with permuted tables gave models explaining less than 1% of the variance, showing that the performance of the models is not due to overfitting.”*

Regarding model selection, the reviewer correctly points out that this is an important aspect of mathematical modeling. In the manuscript, we analyze and describe a sequence of models, with complexity ranging from mononucleotide dependence at a single base to all-by-all base-base interactions. The reason we did this was to give the reader a clear understanding of the tradeoff between model complexity and predictive performance. We therefore argue that model model selection is exactly what we have done. An alternative approach (which the reviewer seems to suggest) would be to introduce (for example) L_1 -penalization in the all-by-all models, vary the scale, and track how the model changes. However, we decided against this approach as we want the paper to be accessible to as wide as an audience as possible, and we since were concerned that using this approach would make a paper that already is quite technical out-of-reach to some readers.

Finally, the reviewer suggests that we work with the Rohs lab to analyze their raw, unpublished, Monte-Carlo data. While this will indeed be very interesting for a future research project, the goal in the current manuscript is to give the reader a good understanding of the shape pentamer tables that have been published and that are widely used by the community.

“7. In several places (Figs 2 and 5, and related text) the authors made statements about shape-features in general, but showed results only on MGW or ProT. It is not clear why the authors did not show results from the other features.”

The reason we only included MGW and ProT in the original manuscript was that the MGW is the most complex shape parameter (in that the sequence dependence spans

5bps) whereas ProT much simpler in that it mostly is an indicator of C/G. We agree with the reviewer that showing all four features is appropriate and the updated Figures 2 and 5 do so.

“8. The idea of shape projection is not clearly conveyed by Figure 5. The authors should consider providing a detailed visualization.”

We have expanded Figure 5 to provide more details.

“9. Providing code would help with reproducibility.”

We agree with the reviewer that reproducibility is important and to address this we have developed a software package that is available on GitHub. See response to reviewer #1 above.

“Typos:

Pg 2, paragraph 1: "and measuring in vitro affinities in high-throughput [3-7]" -> should be "... high-throughput assays ..."

Pg 5, paragraph 1: "minor grove"  "minor groove"

Pg 5, paragraph 2: "that complete neglects"  "that completely neglects"

Pg 13: "FRD"  FDR

Pg 16: "DNA shape shape profiles"  shape twice

Pg 18: "Given an mechanism agnostic"  "a mechanism agnostic"

Pg 19: "When probability the distribution Q  " "when the probability distribution Q" “

Again, we thank the reviewer for taking the time to read the manuscript with great care and for catching these typos. They have all been corrected.

Thank you for sending us your revised manuscript. We have now heard back from the two referees who were asked to evaluate your study. As you will see below, reviewer # 3 still lists some remaining concerns, which we would ask you to address in a revision.

The most substantial remaining concern of reviewer #3 is that further analyses (e.g. on data from another TF family) need to be included to more convincingly support the generality of the main conclusions. During our pre-decision cross-commenting process, in which the referees are given the chance to comment on each other's reports, reviewer #1 agreed with this issue raised by reviewer #3 and mentioned: "I am reasonably convinced by the method but I was a bit disappointed that the authors did not apply it to a larger number of TFs. So I think it is a legitimate request to apply it to at least one more TF family." As such, we would ask you to include in your revised manuscript some additional analyses in this direction.

REVIEWER REPORTS

Reviewer #1:

I am satisfied by the authors' response and have no further comments.

Reviewer #3:

In this revised manuscript Rube et al. gave a much clearer description of their claims and methodology, and added analyses of MAX and CEBPB. However, as discussed below, it is still questionable whether the conclusions will hold in general. Major concerns and suggestions are given below.

1. The section "Direct modeling of shape readout" still seems to be based only on the Exd-Scr dataset. Generality of the conclusion "most of the shape readout is implicitly encoded in TF binding models that only use mono- and di-nucleotides as predictors" is therefore questionable. The authors should either expand the analysis (see below) or carefully restate such concluding remarks.
2. It is also important to clarify that the observed effectiveness of the proposed shape-projection approach is contingent on the current data published from Remo Rohs' Lab and is based on the four common features. The conclusions may not necessarily hold for other features and/or data generated from alternate/improved methods.
3. Re: expanding the dataset, the authors should apply their methods at least on one family of TFs (e.g., from the dataset modeled by Yaron et al., MSB 2017), and show how and whether shape-projection can discover different shape-readouts by those TFs. Such different shape-readouts can then be compared side by side with Yaron et al.'s shape-PWMs and this will provide an informative comparison between direct modeling versus the post-hoc analysis approach.
4. Shape projection is defined on Pg 23 as "Given a mechanism-agnostic mono- and di-nucleotide binding model, shape projection is a post-processing step that identifies the mononucleotide-plus-shape model that best approximates the mechanism-agnostic model". This requires a model constructed from mono- and di-nucleotide features. As such, the discussion on Pg 11 that "An advantage of our two-step approach is that ... the post hoc analysis can be performed without revisiting the underlying data" does not seem to hold - the public databases do not have di-nucleotide models, and the authors did not show how effective shape-projection is if the post-hoc analysis is performed only on mono-nucleotide PWMs.

Minor comments:

1. The authors noted in the rebuttal that "it is not clear to us why leave-one-out cross validation (leaving out reverse-complement pairs) would not fully control for possibility of overfitting". The reviewer's comment was based on the well-known caveat of LOOCV that it can yield models with high-variance, which can result from overfitting the data. Without understanding the data better, it is difficult to say whether a parameter-rich model's validation based on LOOCV is sufficient.
2. Typos:
 - a) Pg 11. "... can be attribute to ..." -> "... attributed to ..."

- b) Pg 3. "improves to ability to predict TF binding in vivo (Zhou et al) and in vitro (Mathelier et al)," -> "improves the ability". Also, Zhou et al used in vitro and Mathelier et al used in vivo data.
- c) Pg 20. "others have inferred binding TF binding models" -> "inferred TF binding models"

2nd Revision - authors' response

26 January 2018

Reviewer #1:

“I am satisfied by the authors' response and have no further comments.”

We are glad that the reviewer is satisfied with our response.

Reviewer #3:

“In this revised manuscript Rube et al. gave a much clearer description of their claims and methodology, and added analyses of MAX and CEBPB. However, as discussed below, it is still questionable whether the conclusions will hold in general. Major concerns and suggestions are given below”

We are happy that the reviewer appreciated our revised manuscript. The remaining concerns are all addressed below.

“1. The section "Direct modeling of shape readout" still seems to be based only on the Exd-Scr dataset. Generality of the conclusion "most of the shape readout is implicitly encoded in TF binding models that only use mono- and di-nucleotides as predictors" is therefore questionable. The authors should either expand the analysis (see below) or carefully restate such concluding remarks.”

The reason we only analyze the Exd-Scr dataset in the section “*Direct modeling of shape readout*” is that we do not advocate using this method but instead advocate using the shape projection method introduced later in the manuscript. We have moderated the concluding statement of the paragraph to read “*The resulting model performs virtually identically to that obtained using the original MGW table ($r^2=0.789$, $p=0.788$, Figure 4d), consistent with the above observation that over 90% of the shape variance in the pentamer tables can be accounted for without including trinucleotide features.*”

“2. It is also important to clarify that the observed effectiveness of the proposed shape-projection approach is contingent on the current data published from Remo Rohs' Lab and is based on the four common features. The conclusions may not necessarily hold for other features and/or data generated from alternate/improved methods.”

While this manuscript was in review, the Rohs lab published pentamer tables for additional DNA shape parameters as well as the electrostatic potential (Li. et al. PMID 29165643, Chiu et al, PMID 29040720). Extending the analysis with these new features is indeed an interesting future research direction and in the revised manuscript we highlight this towards the end of the conclusion: “*While this manuscript was in review, two studies generated pentamer lookup tables for additional DNA shape parameters and for the electrostatic potential in the minor groove (Chiu et al, 2017; Li et al, 2017). It should be straightforward to use our Shapelizer software to analyze readout of these additional structural features.*”

“3. Re: expanding the dataset, the authors should apply their methods at least on one family of TFs (e.g., from the dataset modeled by Yaron et al., MSB 2017), and show how and whether

shape-projection can discover different shape-readouts by those TFs. Such different shape-readouts can then be compared side by side with Yaron et al.'s shape-PWMs and this will provide an informative comparison between direct modeling versus the post-hoc analysis approach.”

We agree with the reviewer that expanding the analysis to additional TF families would be informative for the reader. NRLB is a new computational method and we have not yet used it to reanalyze all publicly available SELEX-seq data. However, to address the reviewer’s concern we analyzed a representative set of additional TFs for which high-quality SMiLE-seq binding data is available and for which NRLB gives robust single-binding mode model fits. Specifically, we extended the demonstration of the shape-projection method to the TFs Bcd (homeodomain), GABP α (ETS-family), GR (nuclear receptor family), SP4 (Sp1 family zinc finger), YY1 (GLI-Kruppel class of zinc finger), and ZEB1 (zinc-finger class homeodomain). Of these, GR, SP4, YY1, and ZEB1 do not have shape sensitivity coefficients that are statistically significant, which highlights the conservative nature of our approach.

The statistically significant shape sensitivity coefficients we found are for the MGW in Bcd and GABP α . For Bcd, the structure has been solved using NMR spectroscopy and it was found that the N-terminal arm of Bcd is unstructured and makes contacts with the minor groove (like in the Exd-Scr complex). This contact is close to the basepair identified by the shape projection. The structure of GABP α bound to DNA has only been solved in the context of the GABP α -GABP β heterodimer complex. Inspection of this structure revealed that a leucine residue in the loop between α helix one and two is located in the minor groove close the base pair identified by the shape projection. In the revised manuscript we speculate that these observations could rationalize the shape readout identified using the shape projection method.

These new results have been added as panels in main Figure 5 and supporting Figure EV5.

Unfortunately, comparing the above results with Yang & Orenstein et al. is not straightforward for a number of reasons:

- 1) For Bcd and GABP α the readout is not available in Yang & Orenstein.
- 2) For AR and MAX we do not find any significant readout and, consistent with this, Yang & Orenstein find “shape-PWM” values that appear to be small. However, this importance of similarity is difficult to judge since the “shape-PWMs” do not have directionality (is narrow or wide minor groove preferred?), since there is no filtering for statistically significance readout in Yang & Orenstein, and since it is unclear how to put the Δr^2 value in Yang & Orenstein on the same scale as our $-\Delta\Delta G/RT$ values.
- 3) For C/EBP β we find a clear preference for heterodimer binding in the HT-SELEX data with NRLB, whereas Yang & Orenstein et al. appear to have identified a monomer binding mode, making any comparison tenuous.

“4. Shape projection is defined on Pg 23 as "Given a mechanism-agnostic mono- and di-nucleotide binding model, shape projection is a post-processing step that identifies the mononucleotide-plus-shape model that best approximates the mechanism-agnostic model". This requires a model constructed from mono- and di-nucleotide features. As such, the discussion on

Pg 11 that "An advantage of our two-step approach is that ... the post hoc analysis can be performed without revisiting the underlying data" does not seem to hold - the public databases do not have di-nucleotide models, and the authors did not show how effective shape-projection is if the post-hoc analysis is performed only on mono-nucleotide PWMs."

We recognize the reviewer's concern. We have removed the sentence *"which is an advantage when underlying data is not readily available or impractical to reprocess (such as when online databases only curate the scoring matrix)"* and instead added: *"for example when additional shape parameters become available (Chiu et al, 2017; Li et al, 2017)"*.

"Minor comments:

1. The authors noted in the rebuttal that "it is not clear to us why leave-one-out cross validation (leaving out reverse-complement pairs) would not fully control for possibility of overfitting". The reviewer's comment was based on the well-known caveat of LOOCV that it can yield models with high-variance, which can result from overfitting the data. Without understanding the data better, it is difficult to say whether a parameter-rich model's validation based on LOOCV is sufficient."

We appreciate the review's point. While the prediction error estimated using LOOCV has smaller bias than that estimate using (for example) 5-fold CV, the variance of the LOOCV estimate can be larger, which potentially could lead to suboptimal selection of predictors, and to suboptimal performance when applying a model to unseen data.

To address this, we reran the analysis using 5-fold cross validation. For the mononucleotide model this gave the r^2 values 0.601443, 0.668934, 0.742224, and 0.712011 for MGW, ProT, Roll, and HelT, respectively, compared to 0.602527, 0.668300, 0.743934 and 0.710127 for LOOCV. For the dinucleotide 0.920353, 0.964242, 0.986408, and 0.980561 for 5-fold CV compared to 0.919934, 0.965057, 0.986530, 0.980431 for LOOCV. The difference between the LOOCV estimate and the 5-fold CV estimate is thus of order 10^{-3} , which does not affect any of the observations in the paper.

"2. Typos:

a) Pg 11. "... can be attribute to ..." -> "... attributed to ..."

b) Pg 3. "improves to ability to predict TF binding in vivo (Zhou et al) and in vitro (Mathelier et al)," -> "improves the ability". Also, Zhou et al used in vitro and Mathelier et al used in vivo data.

c) Pg 20. "others have inferred binding TF binding models" -> "inferred TF binding models"

We thank the reviewer reading the manuscript with great care and for catching these typos. We have corrected them.

Thank you again for sending us your revised manuscript. We are now satisfied with the modifications made and I am pleased to inform you that your paper has been accepted for publication.

Reviewer #1:

I am satisfied with the response.

Corresponding Author Name: Harmen Bussemaker

Manuscript Number: MSB-17-7902